# Modeling the dynamical sinking of biogenic particles in oceanic flow

Pedro Monroy[1], Emilio Hernández-García[1], Vincent Rossi[1], and Cristóbal López[1]

[1]IFISC, Instituto de Física Interdisciplinar y Sistemas Complejos (CSIC-UIB), 07122 Palma de Mallorca, Spain

*Correspondence to:* Pedro Monroy (pmonroy@ifisc.uib-csic.es)

**Abstract.** We study the problem of sinking particles in a realistic oceanic flow, with major energetic structures in the mesoscale, focussing in the range of particle sizes and densities appropriate for marine biogenic particles. Our aim is to evaluate the relevance of theoretical results of finite size particle dynamics in their applications in the oceanographic context. By using a simplified equation of motion of small particles in a mesoscale simulation of the oceanic velocity field, we estimate the influence of physical processes such as the Coriolis force and the inertia of the particles, and we conclude that they represent negligible corrections to the most important terms, which are passive motion with the velocity of the flow, and a constant added vertical velocity due to gravity. Even if within this approximation three-dimensional clustering of particles can not occur, two-dimensional cuts or projections of the evolving three-dimensional density can display inhomogeneities similar to the ones observed in sinking ocean particles.

## 1 Introduction

The sinking of small particles suspended in fluids is a topic of both fundamental importance and of practical implications in diverse fields ranging from rain nucleation to industrial processes (Michaelides, 1997; Falkovich and Fouxon, 2002).

In the oceans, photosynthesis by phytoplankton in surface waters uses sunlight, inorganic nutrients and carbon dioxide to produce organic matter which is then exported downward and isolated from the atmosphere (Henson et al., 2012), a process which forms the so-called biological carbon pump. The downward flux of carbon-rich biogenic particles from the marine surface due to gravitational settling, one of the key process of the biological carbon pump, is responsible (together with the solubility and the physical carbon pumps) of much of the oceans' role in the Earth carbon cycle (Sabine et al., 2004). Although most of the organic matter is metabolized and remineralized in surface waters, a significant portion sinks into deeper horizons. It can be sequestered on various time scales spanning a few years to decades in central and intermediate waters, several centuries in deep waters and up to millions of years locked up in bottom sediments (DeVries et al., 2012). Suitable modeling of the sinking process of particulate matter is thus required to properly assess the amount of carbon sequestered in the ocean and in general to better understand global biogeochemical cycling and its influence on the Earth climate.

This is a challenging task that involves the downward transport of particles of many different sizes and densities by turbulent ocean flows which contain an enormous range of interacting scales. In the oceanographic community, numerous studies approached this problem by considering biogenic particles transported in oceanic flow as passive particles with an added constant velocity in the vertical to account for the sinking dynamics (Siegel and Deuser, 1997; Siegel et al., 2008; Qiu et al., 2014;

Roullier et al., 2014; van Sebille et al., 2015). They suggest that the sinking of particles may not be strictly vertical but oblique, meaning that the locations where the particles are formed at the surface may be distant from the location of their deposition in the seafloor sediment. Then Siegel et al. (2008) presented the concept of statistical funnels which describe and quantify the source region of a sediment trap (subsurface collecting device of sinking-particles used to get estimate of vertical fluxes). The validity of this approximation and the influence of different physical processes is however poorly discussed in these analyses.

In the physical community, the framework to model sinking particles is based on the Maxey-Riley-Gatignol equation for a small spherical particle moving in an ambient flow (Maxey and Riley, 1983; Gatignol, 1983; Michaelides, 1997; Provenzale, 1999; Cartwright et al., 2010), which highlights the importance of mechanisms beyond passive transport and constant sinking velocity, such as the role of finite size, inertia and history dependence. A major outcome of these studies is that inhomogeneities and particle clustering can arise spontaneously even if the fluid velocity field is incompressible and particles do not interact (Squires and Eaton, 1991). Particle clustering and patchiness is indeed observed in the surface and subsurface of the ocean (Logan and Wilkinson, 1990; Buesseler et al., 2007; Mitchell et al., 2008)

Here we consider the theory of small but finite-size particles driven by geophysical flows, which is, as mentioned above, conveniently based on the Maxey-Riley-Gatignol equation. In Sect. 2 we review the main characteristics of marine particles which are relevant for their sinking dynamics. In Sect. 3 we present the equations of motion describing this process, together with the approximations required to obtain them and the type of particles for which they are valid. In particular, we discuss its validity and the relevance of the different physical processes involved in the range of sizes and densities of marine biogenic particles. In Sect. 4 we use these equations to study the settling dynamics in a modelled oceanic velocity field produced by a realistic high-resolution regional simulation of the Benguela upwelling system (southwest Africa). We estimate the relevance of physical processes such as the Coriolis force and the inertia of the particles with respect to the settling velocity. We also observe the spatial distribution of particles falling onto a plane of constant depth above the seabed and we identify clustering of particles that is interpreted with simple geometrical arguments which do not require physical phenomena beyond passive transport and constant terminal velocity. Our main results are finally summarized in a Conclusion section.

## 2   Characteristics of marine biogenic particles

In theory, the sinking velocities of biogenic particles depend on various intrinsic factors (such as their sizes, shapes, densities, porosities) which can be modified along their fall by complex bio-physical processes (e.g. aggregation, ballasting, trimming by remineralisation) as well as by the three-dimensional flow field (Stemmann and Boss, 2012). However reasonable estimates of the effective sinking velocities of marine particles can be obtained by taking into account only its size and density (McDonnell and Buesseler, 2010). In our Lagrangian setting we thus consider that the two key properties of marine particles controlling their sinking dynamics are their size and density. Here we present the standard classification of marine particles according to the typical range of size and density by compiling different bibliographical sources.

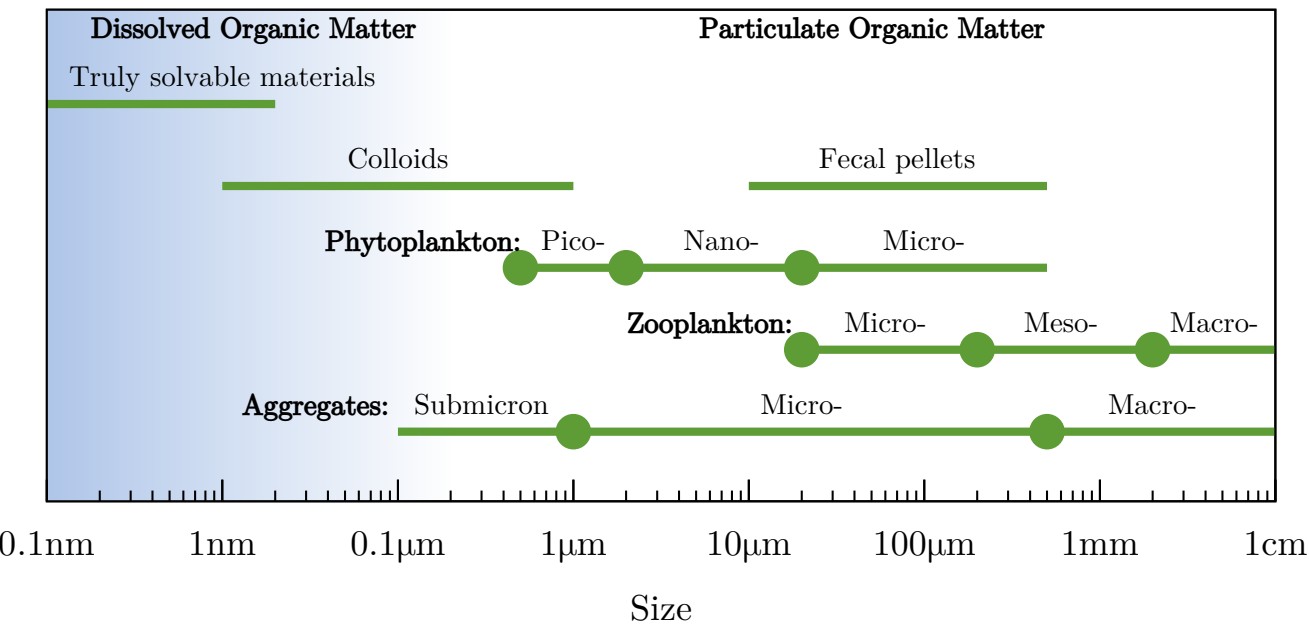

**Figure 1.** Size and classification of marine particles (adapted from Simon et al. (2002)).

## 2.1 Size

Because of the diversity of the shapes, the size of a particle refers to the diameter of a sphere of equivalent volume (Equivalent Spherical Diameter) (Guidi et al., 2008). The size of marine particles ranges from $1\ nm$ (almost-dissolved colloids) to aggregates larger than $1\ cm$ (Stemmann and Boss, 2012).

5    Originally, the size classification of particles was based on the minimal pore size of the nets used for their collection, which is about $\simeq 0.45 - 1.0\ \mu m$. Any material larger than $0.2\ \mu m$ (thus isolated by the filtration of seawater) is regarded as particulate organic matter, while the fraction that percolates through the filter is labelled as dissolved matter. This includes colloidal and truly dissolved materials (see Fig. 1). Although this discrimination of the size-continuum observed in the real ocean is somehow arbitrary, it is useful –and we will follow it– because particles smaller than $1.0\ \mu m$ are not prone to sinking (Hedges, 2002).

10    In the following, our focus is thus on particulate matter larger than $1.0\ \mu m$ (Fig. 1). Organic matter is produced in the sun-lit layer of the ocean by the primary production through photosynthesis of autotrophic microbes (mainly bacteria and phytoplankton). During their lifetime growth they exude colloidal and small particles to finally form larger particles when they

die. Dead phytoplankton are within the range of 1 $\mu m$ (picoplankton, e.g. cyanobacteria) and a few hundred of micrometers (microphytoplankton, e.g. diatoms).

Thereafter zooplankton consumes alive phytoplankton and inert particles and produce fecal pellets and dead bodies. Most fecal materials have enough size to sink rapidly by their own (De La Rocha and Passow, 2007). Typical sizes of such particles are 10 $\mu m$ for a pellet of copepod of 200 $\mu m$ length (Jackson, 2001), krill fecal pellets are between 160 $\mu m - 460$ $\mu m$ (McDonnell and Buesseler, 2010) and euphausiid fecal pellets span 300 $\mu m - 3$ $mm$ (Komar et al., 1981), providing the total range of $10 \mu m$ to 3 $mm$. Concerning the zooplankton dead bodies, they are divided in micro-, meso- and macro-, with sizes in the range $20 \mu m - 1cm$. A detailed summary is given in Table 1.

Finally, there are the so-called organic aggregates which occur in the size range of $1 \mu m$ to $10cm$. They are typically formed *in-situ* by physical aggregation or biological coagulation and are usually composed of numerous planktonic individuals and fecal pellets sticked together within a colloidal matrice. They are often distinguished in three size classes (Simon et al., 2002): macroscopic aggregates or macro-aggregates $> 5mm$ usually called marine snow; microscopic, from 1 to $500 \mu m$, also known as micro-aggregates; and submicron particles $< 1 \mu m$ (which do not sink).

## 2.2 Density

The density of marine particles depends on their composition which can be divided into a mineral and a organic fraction (Maggi and Tang, 2015). The mineral or inorganic matter consists of biogenic minerals: Particulate Inorganic Carbon (PIC), e.g. calcium carbonate produced by coccoliths with density 2700 $kg/m^3$ and Biogenic Silica (BSi), produced by diatoms, significantly less denser than PIC, 1950 $kg/m^3$ (Balch et al., 2010). The density of Particulate Organic Matter (POC) ranges widely depending on its origin. For instance, the density of cytoplasm spans from 1030 to 1100 $kg/m^3$, while the one of fecal pellets ranges between 1174 $kg/m^3$ and 1230 $kg/m^3$ (Komar et al., 1981). Despite this variability, it is possible to assign a range to the density of organic matter, from 1050 to 1500 $kg/m^3$.

Considering all these estimates together, the density of marine particle ranges approximately between 1050 to 2700 $kg/m^3$ (Maggi, 2013). This should be compared a standard value for sea water density in the interior ocean which spans roughly 1020-1030 $kg/m^3$. Thus most of the particle types described previously will sink. Assuming constant size and density for each particle along its downward course, we deduce that most of the particles types described previously will sink. This holds without considering biogeochemical and (dis)aggregation processes that may occur in nature, thus lowering the particle density and resulting in clustering and trapping of particles at particular isopycnals (Sozza et al., 2016). Note that we do not consider here living organisms which show vertical movements by active swimming or by controlling their buoyancy (Moore and Villareal, 1996; Azetsu-Scott and Passow, 2004).

| Individual Particles (mostly organic) | Aggregates (compounds of organic and inorganic particles) |
|---|---|
| Fecal pellets (cylindrical):<br><br>  – Krill fecal pellets: Length between $400 \ \mu m$ and $9 \ mm$, diameter $120 \ \mu m$ (McDonnell and Buesseler, 2010). ESD ($160 \ \mu m - 460 \ \mu m$)<br><br>  – $10 \ \mu m$, consistent with pellet volume of a $200 \ \mu m$ copepod (Jackson, 2001)<br><br>Dead zooplankton (Stemmann and Boss, 2012):<br><br>  – Macrozooplankton:<br>   size $> 2000 \ \mu m$<br><br>  – Mesozooplankton:<br>   $200 <$ size $< 2000 \ \mu m$<br><br>  – Microzooplankton:<br>   $20 <$ size $< 200 \ \mu m$<br><br>Dead phytoplankton (Stemmann and Boss, 2012):<br><br>  – Microphytoplankton:<br>   (size$> 200 \ \mu m$)<br><br>  – Nanophytoplankton:<br>   ($20 <$size$< 200 \ \mu m$)<br><br>  – Picophytoplankton:<br>   ($2 <$ size $< 20 \ \mu m$) | Aggregates(Simon et al., 2002):<br><br>  – Macroscopic (Marine Snow):<br>   size $> 500 \ \mu m$.<br><br>  – Microscopic:<br>   $1 \mu m <$ size $< 500 \ \mu m$.<br><br>  – Submicron:<br>   size $< 1 \ \mu m$. |

**Table 1.** Simplified categorization of marine biogenic particles, and their associated sizes.

## 3 Equations of motion for small spherical rigid particles

### 3.1 The Maxey-Riley-Gatignol equation

To describe the sedimentation of biogenic particles, we need to study the motion of single particles driven by fluid flow. A milestone to analyze the dynamics of a small spherical rigid particle of radius $a$ subject to gravity acceleration $\mathbf{g}$ in an unsteady fluid flow $\mathbf{u}(\mathbf{r}, t)$ is given by the Maxey-Riley-Gatignol (Maxey and Riley, 1983; Gatignol, 1983; Michaelides, 1997; Cartwright et al., 2010) equation (MRG in the following):

$$
\begin{aligned}
\rho_p \frac{d\mathbf{v}}{dt} =& \rho_f \frac{D\mathbf{u}}{Dt} + (\rho_p - \rho_f)\mathbf{g} - \frac{9\nu\rho_f}{2a^2}\left(\mathbf{v} - \mathbf{u} - \frac{a^2}{6}\nabla^2\mathbf{u}\right) \\
& - \rho_f\left(\frac{d\mathbf{v}}{dt} - \frac{D}{Dt}(\mathbf{u} + \frac{a^2}{10}\nabla^2\mathbf{u})\right) \\
& - \frac{9\rho_f}{2a}\sqrt{\frac{\nu}{\pi}}\int_0^t \frac{\frac{d}{ds}(\mathbf{v} - \mathbf{u} - \frac{a^2}{6}\nabla^2\mathbf{u})}{\sqrt{t-s}}ds.
\end{aligned} \tag{1}
$$

The velocity of the particle is denoted by $\mathbf{v} = \mathbf{v}(t)$. The particle and fluid densities are $\rho_p$ and $\rho_f$, respectively, and $\nu$ denotes the fluid kinematic viscosity. The time derivative operators $\frac{d}{dt} = \frac{\partial}{\partial t} + \mathbf{v} \cdot \nabla$ and $\frac{D}{Dt} = \frac{\partial}{\partial t} + \mathbf{u} \cdot \nabla$ denote the time rate of change following the particle itself and the time rate of change following a fluid element in the undisturbed flow field $\mathbf{u}(\mathbf{r}, t)$ respectively. This equation of motion gives the balance between the different forces acting on the particle, which corresponds to the right-hand-side terms: the pressure force (the force exerted on the particle by the undisturbed flow), the buoyancy force, the drag force (Stokes drag), the added mass force resulting from the part of the fluid moving with the particle, and the history force. As will be discussed below the validity of this equation requires several conditions, being the main one the small size of the particles. The terms with $a^2 \nabla^2 \mathbf{u}$ are the Faxén corrections (Faxén, 1922).

The full MRG is very complicated to manage. A further simplification is usually performed based on the single assumption of very small particles (what this exactly means will be discussed later on). With this, the Faxén corrections and, as commented below, also the history term (since $a/\sqrt{\nu} << 1$) can be neglected (Maxey and Riley, 1983; Michaelides, 1997; Haller and Sapsis, 2008). Note however that the history term can be relevant under some conditions, as for example larger particle size (Daitche and Tél, 2011; Guseva et al., 2013, 2016; Olivieri et al., 2014). Thus we obtain the standard form of the MRG equations (Maxey and Riley, 1983):

$$\frac{d\mathbf{v}}{dt} = \beta \frac{D\mathbf{u}}{Dt} + \frac{\mathbf{u} - \mathbf{v} + \mathbf{v}_s}{\tau_p}, \tag{2}$$

where $\beta = \frac{3\rho_f}{2\rho_p + \rho_f}$, the Stokes time is $\tau_p = \frac{a^2}{3\beta\nu}$, and $\mathbf{v}_s = (1 - \beta)\mathbf{g}\tau_p$ is the settling velocity in quiescent fluid. Equation (2) is the starting point for most inertial particle studies (Michaelides, 1997; Balkovsky et al., 2001; Cartwright et al., 2010).

We now discuss the validity of the MRG equation Eq. (1) or rather its simplified form Eq. (2) for the range of sizes and densities of marine organisms. We do so in the context of open-ocean flows, which are typically most energetic at the mesoscale (scales of about 100 km), and where there is a strong stratification, with vertical velocities three or four orders of magnitude smaller than horizontal ones. The motion becomes more three-dimensional, and then the concepts of three-dimensional turbulence more relevant, below scales $l$ of some hundred of meters, with typical velocities decreasing as $l^{1/3}$ for decreasing scale and velocity gradients increasing as $l^{-2/3}$ until the Kolmogorov scale $l = \eta$ below which flow becomes smooth. Because of its direct exposure to wind, turbulence intensity is typically larger at the ocean surface, with values of turbulent energy dissipation in the range $1 \cdot 10^{-6} m^2/s^3 < \epsilon < 3 \cdot 10^{-5} m^2/s^3$ (Jimenez, 1997), than at depth. The first condition for the validity of the MRG equation that was originally discussed by Maxey and Riley (1983) is that the particles have to be much smaller than the typical length scale of variation of the flow. This means that for multiscale (turbulent) flows the radius of the particle $a$ has to be much smaller than the Kolmogorov scale $\eta$, which according to the previous values of $\epsilon$, is typically $0.3mm < \eta < 2mm$ in the ocean surface (Okubo, 1971; Jimenez, 1997). Note that we only have to consider worst-case situations for assessing the validity of the different approximations. Another condition to be fulfilled is that the shear Reynolds number must be small $Re_\nabla = a^2 U/\nu L << 1$, where $U$ and $L$ are typical velocity and length scales. For a turbulent ocean with multiple scales and velocities, the most restrictive condition arises when they take the values of the Kolmogorov velocity $v_\eta$ and length $\eta$, respectively, since then the velocity gradients are maxima. In this case the condition becomes $Re_\nabla = a^2/\eta^2 << 1$, which again is satisfied for small particles. We note that Guseva et al. (2013) found that the relative importance of the history term in Eq. (1)

with respect to the drag force is of the order of a parameter which in our notation is $(Re_\nabla)^{1/2}$. This justifies neglecting the history term for small particles, although its importance increases for increasing size (Daitche and Tél, 2011; Guseva et al., 2013).

Another condition to be satisfied for the validity of the MRG equation is that the so-called Reynolds particle number, $Re_p = \frac{a|\mathbf{v}-\mathbf{u}|}{\nu}$ should fulfill $Re_p << 1$. Considering that gravity force dominates over other forces one has $|\mathbf{v}-\mathbf{u}| \simeq |\mathbf{v}_s| \equiv v_s$, where $\mathbf{v}_s$ is, as introduced before, the settling velocity of particles in a quiescent fluid due to Stokes drag. The Reynolds particle number is then $Re_p = \frac{av_s}{\nu}$. Note that the settling velocity depends only on the densities of particles via the parameter $\beta$. Assuming a mean density of sea water in the upper ocean as $\rho_f = 1025 kg/m^3$ the parameter $\beta$ has values within the range $[0.5, 0.99]$ for the typical values of the density of marine particles previously discussed. Fig. 2 shows $v_s$ for different sizes and the regions where $Re_p > 1$ (and other parameter regions where MRG is not a good approximation) as a function of particle radius and for the limiting values of $\beta$. It reveals that Eq. (1) can not describe ocean particles larger than $300\mu m$ of any density, and for a limited range of densities when the particle radius exceeds approximately $100\mu m$. In fact, the range of application of MRG to marine particles is plotted in the blue area, which at the same time gives an estimate of the typical sinking velocities for a given particle size.

Summarizing, both the MRG and its approximation Eq. (2) are valid for marine particles with size within the range $1\mu m$ and $200\mu m$. That is, it is valid for all particulate organic matter in Fig. 1 except the largest of the micro-aggregates and meso- and macro-bodies of zooplankton. The sinking velocities range from $1mm/day$- $1km/day$.

## 3.2 The MRG equation in a rotating frame and further simplications

We are interested in applying Eq. (2) in oceanic flows, where the particle $\mathbf{v}$ and flow $\mathbf{u}$ velocities are expressed in a frame rotating with the Earth angular velocity $\mathbf{\Omega}$ (Elperin et al., 2002; Biferale et al., 2016; Tanga et al., 1996; Provenzale, 1999; Sapsis and Haller, 2009). Both time derivatives $\frac{d}{dt}$ and $\frac{D}{Dt}$ have to be corrected following the rule

$$\frac{d}{dt} \rightarrow \frac{d}{dt} + 2\mathbf{\Omega} \times \mathbf{v} + \mathbf{\Omega} \times (\mathbf{\Omega} \times \mathbf{r}), \tag{3}$$

$$\frac{D}{Dt} \rightarrow \frac{D}{Dt} + 2\mathbf{\Omega} \times \mathbf{u} + \mathbf{\Omega} \times (\mathbf{\Omega} \times \mathbf{r}). \tag{4}$$

Where $\Omega = |\mathbf{\Omega}|$ and $\mathbf{r}$ is the particle position vector whose origin is in the rotation axis. So that Eq.(2) is now

$$\frac{d\mathbf{v}}{dt} = \beta \frac{D\mathbf{u}}{Dt} - \frac{\mathbf{v}-\mathbf{u}}{\tau_p} - 2\mathbf{\Omega} \times (\mathbf{v}-\beta\mathbf{u}) + \mathbf{v}'_s/\tau_p. \tag{5}$$

Two apparent forces arise in the equation, the Coriolis force $2\mathbf{\Omega} \times (\mathbf{v}-\beta\mathbf{u})$ and the centrifugal force, which is included in a modified sinking velocity $\mathbf{v}'_s = (1-\beta)(\mathbf{g}-\mathbf{\Omega} \times (\mathbf{\Omega} \times \mathbf{r}))\tau_p$. The effect of the centrifugal force is very small (of order $10^{-3}$ compared to gravity) and can be absorbed in a redefinition of $\mathbf{g}$. Thus, in the following we take $\mathbf{v}'_s = \mathbf{v}_s$ with the properly chosen $\mathbf{g}$.

The ratio between the particle response time and the Kolmogorov time scale is the Stokes number $St = \tau_p/\tau_\eta$, which measures the importance of particle's inertia because of its size and density. According to the range of $\epsilon$ in the ocean mentioned

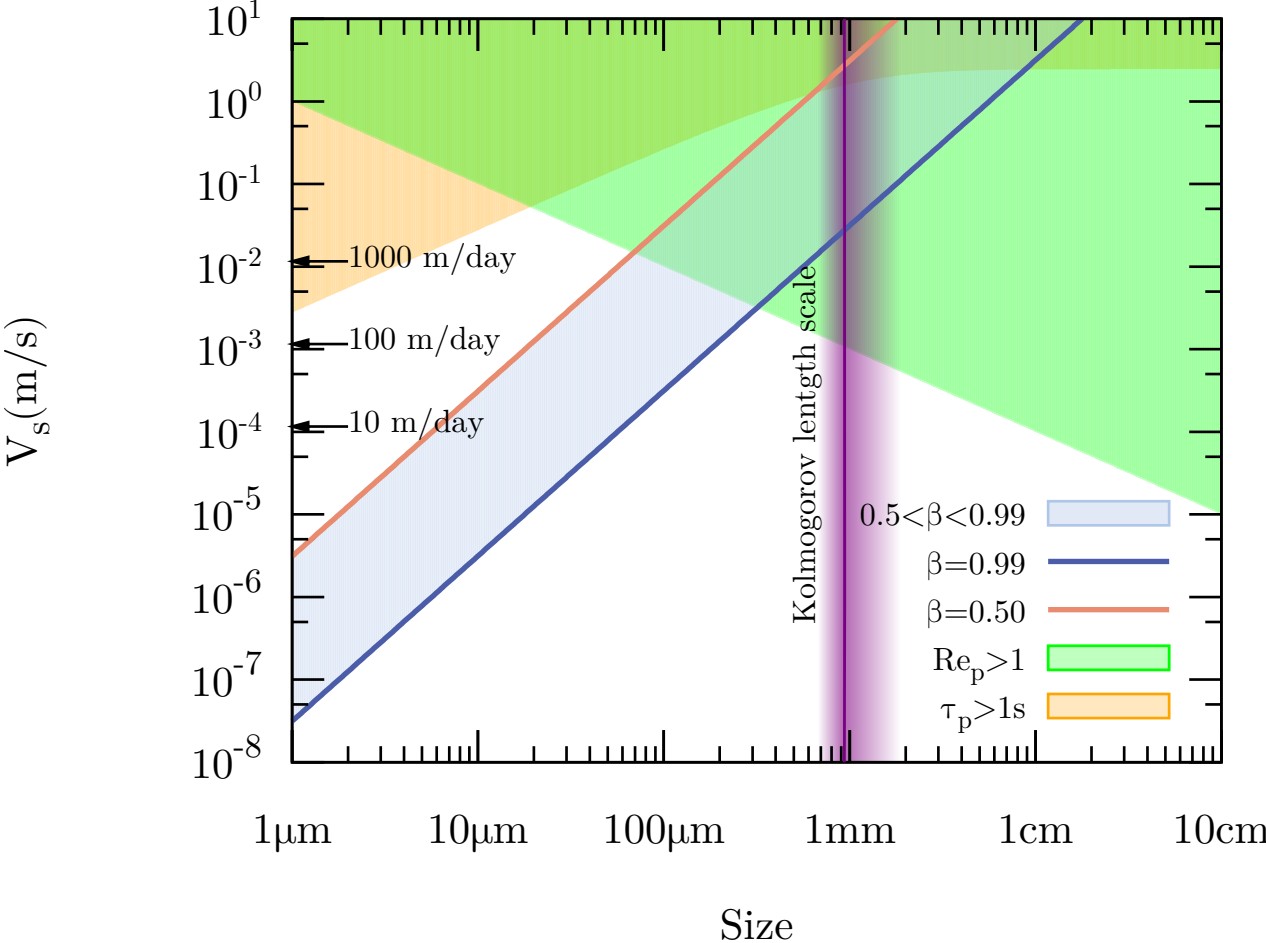

**Figure 2.** Sinking velocity versus particle radius for different $\beta$, which is determined by densities. The blue zone determines the values of the settling velocities at a given radius, as determined by the typical marine particle densities. The green area is determined by the condition $Re_p > 1$ for which the MRG equation is not valid. Use of the MRG equation is also unjustified for particles larger than the Kolmogorov length scale also plotted in the figure. We also show the region $\tau_p > \tau_\eta \approx 1s$ where the additional approximation leading to Eq. (6) becomes invalid.

before, we get $0.1\ s < \tau_\eta < 5\ s$, and for our range of particle sizes $10^{-6}\ s < \tau_p < 10^{-2}\ s$ so we can assume that $St << 1$ (see Fig. 2). This motivates us to make a second (standard) approximation (Balkovsky et al., 2001; Haller and Sapsis, 2008) of the MRG equation expanding in powers of $\tau_p$ (note that it would be more natural to make the expansion in powers of the non-dimensional $St$ but we prefer to do it in $\tau_p$ to control on the time scales of the problem). Assuming first the solution to Eq.

(2):

$$\mathbf{v} = \mathbf{u} + \mathbf{u}_1 \tau_p + \mathbf{u}_2 \tau_p^2 + \dots,$$

and using $\frac{d\mathbf{v}}{dt} = \frac{D\mathbf{u}}{Dt} + O(\tau_p)$, we get that the particle velocity at first order in $\tau_p$ is

$$\mathbf{v} = \mathbf{u} + \mathbf{v}_s + \tau_p(\beta - 1)\left(\frac{D\mathbf{u}}{Dt} + 2\boldsymbol{\Omega} \times \mathbf{u}\right). \tag{6}$$

It is worth recalling that $\tau_p(1-\beta) = v_s/g$, so that all dependencies on particle size and density appear in Eq. (6) through the combination of parameters defining $v_s$. Different combinations of size and density, taken within the ranges reported in Sect. 2, follow the same dynamics if they have the same undisturbed settling velocity $v_s$.

A further discussion of Eq. (6) follows. At this order only three physical processes correct the particle velocity with respect to the fluid velocity: the Stokes friction determining the settling velocity $v_s$, the inertial term given by $\tau_p(\beta-1)\frac{D\mathbf{u}}{Dt}$ whose
major effect is to introduce a centrifugal force pulling particles away from vortex cores (Maxey, 1987; Michaelides, 1997), and the influence of the Coriolis force $2\tau_p(\beta-1)\boldsymbol{\Omega} \times \mathbf{u}$. Concerning sinking dynamics, the $\mathbf{v} = \mathbf{u} + \mathbf{v}_s$ is the most relevant approximation, and many other studies consider it, mainly in oceanographic contexts (e.g. Siegel and Deuser, 1997). Note that we can use the right-hand-side of Eq. (6) with $\mathbf{u} = \mathbf{u}(\mathbf{r}, t)$ to define the particle velocity $\mathbf{v}$ as a velocity field in three-dimensional space $\mathbf{v} = \mathbf{v}(\mathbf{r}, t)$. If one uses the lowest-order approximation $\mathbf{v} \approx \mathbf{u}$ we have $\nabla \cdot \mathbf{v} = \nabla \cdot \mathbf{u} = 0$ when the fluid
velocity field $\mathbf{u}$ is incompressible (which is the case for ocean flows). This means that when considering this term alone, one cannot obtain a compressible particle velocity whereas this was the main reason invoked to explain the clustering of finite-size particles (Squires and Eaton, 1991; Bec, 2003). For this reason, numerous studies (Tanga et al., 1996; Michaelides, 1997; Bec et al., 2007, 2014; Cartwright et al., 2010; Guseva et al., 2013; Gustavsson et al., 2014; Beron-Vera et al., 2015) consider the role of the additional terms. With them $\nabla \cdot \mathbf{v} = \tau_p(\beta-1)\nabla \cdot (\frac{D\mathbf{u}}{Dt} + 2\boldsymbol{\Omega} \times \mathbf{u}) \neq 0$, and inertia-induced clustering may occur. In
the following sections we address two main questions: a) how relevant for the sinking dynamics are the Coriolis and centrifugal terms?; and b) are they essential ingredients for the clustering of biogenic particles? We will study the relevance of the different terms in Eq. (6) in a realistic oceanic setting.

## 4    Numerical simulations

The velocity flow u of the Benguela region was produced by a regional simulation of a hydrostatic free-surface primitive
equations model called ROMS (Regional Ocean Modelling System). The configuration used here extends from 12°S to 35°S and from 4°E to 19°E (blue rectangle in Fig. 3) and was forced with climatological atmospheric data (Gutknecht et al., 2013). The simulation area extends from $12°S$ to $35°S$ and from $4°E$ to $19°E$ (blue rectangle in Fig. 3). The velocity field data set consists of 2 years of daily averages of zonal ($u$), meridional ($v$) and vertical velocity ($w$) components, stored in a 3D grid with a horizontal resolution of $1/12^o$ and 32 vertical terrain-following levels using a stretched vertical coordinate where the layer
thickness increases from surface/bottom to the ocean interior.

In order to integrate particle trajectories from the velocity in Eq. (6) we interpolate linearly $\mathbf{u}(\mathbf{r}, t)$ from the closest space-time grid points to the actual particle locations. Given the huge disparity between the model resolution and the small particle-sizes

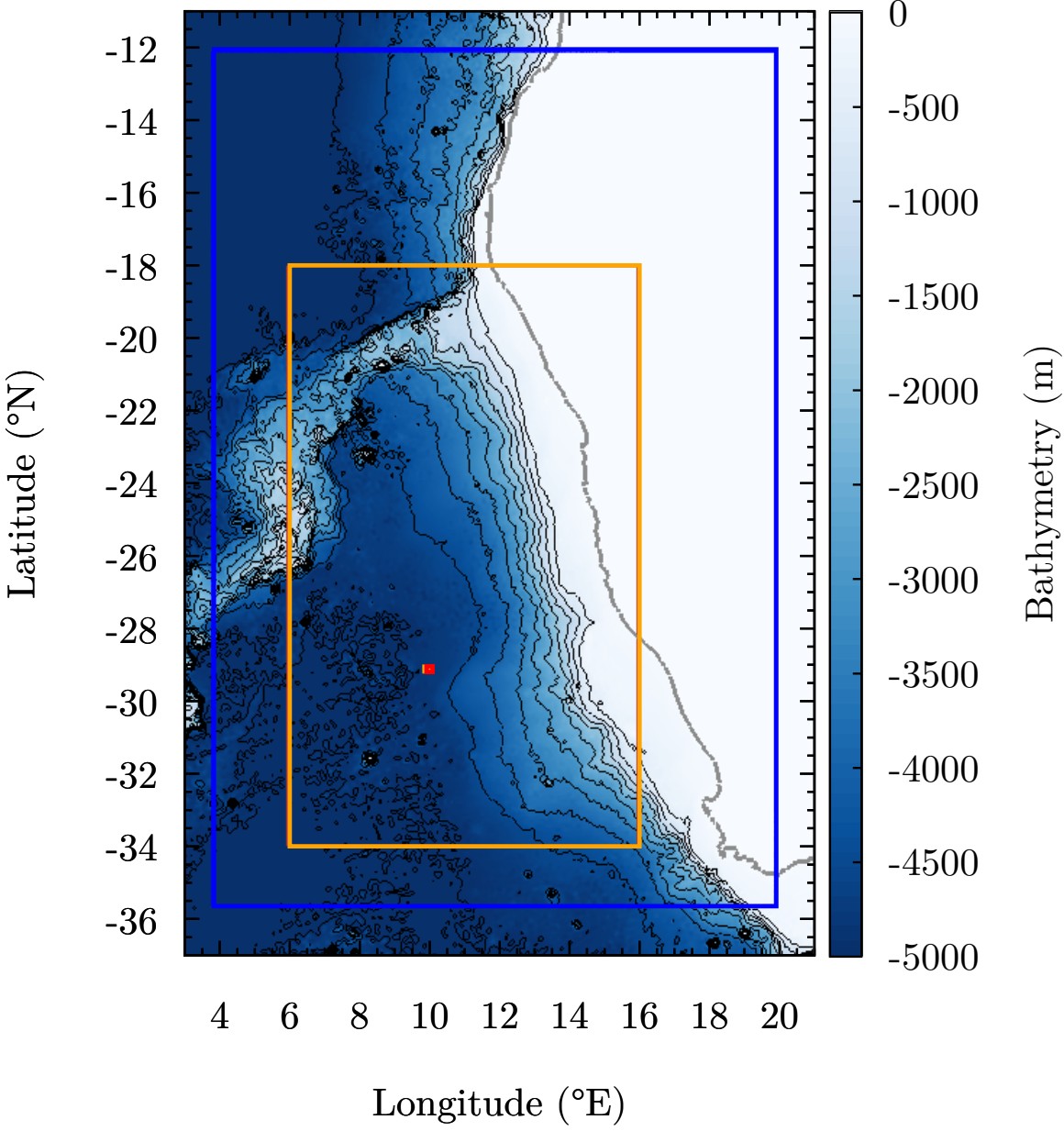

**Figure 3.** Map of region of study. Color corresponds to bathymetry. Blue rectangle is region used for simulations of the ROMS model. Orange rectangle is the region for the clustering numerical experiment of Sect. 5 and red rectangle is the release site of the sinking numerical experiments of Sect. 4. Gray represent the coastline.

considered, it is pertinent to parameterize in some way the unresolved scales. This can be done by different approaches, from stochastic Lagrangian modeling (Brickman and Smith, 2002), to deterministic kinematic fields (Palatella et al., 2014). The first approach is adopted by adding a simple white noise to the particle velocity (Tang et al., 2012), with different intensity in the vertical and horizontal directions. Thus, we consider this noisy version of the simplified MRG:

$$\frac{d\mathbf{r}(t)}{dt} \quad = \quad \mathbf{v}(t) \tag{7}$$

$$\mathbf{v} \quad = \quad \mathbf{u} + \mathbf{v}_s + \tau_p(\beta - 1)\left(\frac{D\mathbf{u}}{Dt} + 2\mathbf{\Omega} \times \mathbf{u}\right) + \mathbf{W}. \tag{8}$$

$\mathbf{W}(t) \equiv \sqrt{2D_h}\mathbf{W}_h(t) + \sqrt{2D_v}W_z(t)$, with $(\mathbf{W}_h, W_z) = (W_x(t), W_y(t), W_z(t))$ a three-dimensional vector Gaussian white noise with zero mean and correlations $\langle W_i(t)W_j(t')\rangle = \delta_{ij}\delta(t - t')$, $i, j = x, y, z$. We consider an horizontal eddy diffusivity, $D_h$, depending on resolution length scale $l$ according to Okubo formula (Okubo, 1971; Hernandez-Carrasco et al., 2011):

$D_h(l) = 2.055 \times 10^4 l^{1.55}$ $(m^2/s)$. Thus, if taking $l \sim 8$ $km = 8000$ $m$ (corresponding to $1/12°$) we obtain $10 m^2/s$. In the vertical direction we use a constant value of $D_v = 10^{-5} m^2/s$ (Rossi et al., 2013).

In order to obtain quantitative assessment of the relative effects of the different physical terms in Eq. (8), we will compare trajectories obtained from the following expressions which only consider some of the terms of the full expression Eq. (8):

$$\mathbf{v}^{(0)} = \mathbf{u} + \mathbf{v}_s + \mathbf{W}, \tag{9}$$

$$\mathbf{v}^{(co)} = \mathbf{u} + \mathbf{v}_s + \tau_p(\beta - 1)2\mathbf{\Omega} \times \mathbf{u} + \mathbf{W}, \tag{10}$$

$$\mathbf{v}^{(in)} = \mathbf{u} + \mathbf{v}_s + \tau_p(\beta - 1)\frac{D\mathbf{u}}{Dt} + \mathbf{W}. \tag{11}$$

Besides the random noise term, the first expression (9) only considers the settling velocity, equation (10) resolves the settling velocity plus the Coriolis effect, and equation (11) considers the settling plus the inertial term.

For the numerical experiments we will consider a set of six values of $v_s$ ranging from $5m/day$ to $200m/day$, with different

integration times to have in all the cases a sinking to about $1000 - 1100$ $m$ depth. The stochastic equation (7) with expressions (8)-(11) is written in spherical coordinates and numerically integrated using a second-order Heun's method with time step of 4 hours (Toral and Colet, 2014). We use $R = 6371$ $km$ for the Earth radius, $g = 9.81m/s^2$, and the angular velocity $\mathbf{\Omega}$ is a vector pointing in the direction of Earth axis and modulus $|\mathbf{\Omega}| = 7.2722 \times 10^{-5}$ $s^{-1}$. We take $v_s$ and $\tau_p$ constant in each experiment because, although water density may increase with depth, this variation is at most of $10kg/m^3$ in the range of depths we are

considering here and then the impact on $v_s$ is below 0.1%. We use as initial starting date 17 September 2008. The numerical experiments consist in launching $N = 6000$ particles from initial conditions randomly chosen in a square of size $1/6°$ centered at $10.0°E$ $29.12°S$ and $-100.0m$ depth (red rectangle in Fig. 3), and in letting them evolve for a given time $t_f$ (stated in Table 2) following Eq. (7) with expressions (8)-(11). We use in each case identical initial conditions and the same sequence of random numbers for the noise terms. In this way we guarantee that any difference in particle trajectories arise from the inclusion or

not of the inertial and Coriolis terms. We obtain the time-dependent positions of all the particles for each approximation to the dynamics: $\mathbf{r}_i(t)$, $\mathbf{r}_i^{(0)}(t)$, $\mathbf{r}_i^{(co)}(t)$, and $\mathbf{r}_i^{(in)}(t)$, $i = 1, ..., N$, following respectively Eqs. (8)-(11) and the corresponding final positions at $t = t_f$.

Table 2 gives the mean and the standard deviation of the depths attained by the set of particles in each numerical experiment as obtained from Eqs. (7) and (8). We find that the use of the different approximations (9)-(11) gives virtually the same results. The only differences larger than $1\ cm$ in mean or standard deviation are the ones for the smallest unperturbed settling velocity considered, $v_s = 5\ m/day$, and are also reported in Table 2. The measured differences are negligible as compared with the

5   traveled distance or even with the model grid size. Indeed small changes in the ROMS model configuration or in the velocity interpolation procedure would have an impact larger than this. The mean displacements in the horizontal obtained with the different approximations (not shown) differ also in less $0.1\%$. We thus conclude that the simplest approximation Eq. (9) which only considers passive transport and an added constant sinking velocity already provides a good description of the sinking process for the type of marine particles and the range of space and time scales considered here. Note that the depth attained by

10   the particles is always slightly shallower than $z = -1100\ m$, which is the depth that would be reached in a still fluid. It is still debated under which conditions fluid flows enhances or reduces the settling velocity (Maxey, 1987; Wang and Maxey, 1993; Ruiz et al., 2004; Bec et al., 2014).

| $v_s$ (m/day) | integration time $t_f$ (days) | Mean final depth (m) | | std final depth (m) | |
|---|---|---|---|---|---|
| 200 | 5 | -1091.78 | | 3.88 | |
| 100 | 10 | -1065.33 | | 6.57 | |
| 50 | 20 | -1033.97 | | 6.22 | |
| 20 | 50 | -1051.85 | | 22.67 | |
| 10 | 100 | -1043.49 | | 51.22 | |
| 5 | 200 | -1054.97 | | 62.03 | |
| | | -1054.76 | (co) | 62.14 | (co) |
| | | -1054.76 | (in) | 62.16 | (in) |
| | | -1054.72 | (0) | 62.14 | (0) |

**Table 2.** Mean and standard deviation of the set of depths attained, according to Eqs. (7) and (8), by the set of particles released from the red rectangle in Fig. 3 at $z = -100\ m$ for the different values of $v_s$ and integration times used. The results labeled $(co)$, $(in)$, and $(0)$ are obtained from the different approximations in Eqs. (9)-(11), which differ more than $1\ cm$ from the ones obtained from Eq. (8) only in the $v_s = 5\ m/day$ case.

We perform now a more stringent test going beyond the analyses of mean displacements by considering differences between individual particle trajectories. To assess the impact of the Coriolis and of the inertial effects we compare the positions $\mathbf{r}_i^{(co)}(t)$,

15   and $\mathbf{r}_i^{(in)}(t)$ with the simpler dynamics $\mathbf{r}_i^{(in)}(t)$ for each time $t$. To do so we compute the root mean square difference in

position per particle and time, which we separate in vertical and horizontal components:

$$r_h^{(k)}(t) = \sqrt{\frac{1}{N}\sum_{i=1}^{N}\left(\mathbf{x}_i^{(0)}(t) - \mathbf{x}_i^{(k)}(t)\right)^2} \tag{12}$$

$$r_v^{(k)}(t) = \sqrt{\frac{1}{N}\sum_{i=1}^{N}\left(z_i^{(0)}(t) - z_i^{(k)}(t)\right)^2} \tag{13}$$

with $\mathbf{x}_i = (x_i, y_i)$, the horizontal position vectors, and the superindex $(k)$ takes the values $(co)$ or $(in)$.

Fig. 4 shows the influence of the Coriolis term in the horizontal component for each sinking velocity as a function of time. We observe an exponential growth in a wide range of times, which reveals the chaotic behavior of each of the compared trajectories. The value of the exponent $0.08 days^{-1}$ is in agreement with the order of magnitude of the Lyapunov exponent calculated using the same ROMS velocity model and region (Bettencourt et al., 2012). Similar exponential growth with the same growth rate were observed for the inertial terms and the vertical components (not shown), although the absolute magnitude of these mean

root square differences was much smaller.

     The horizontal and vertical differences $r_{h,v}^{(co)}$ at the final integration time $t_f$ (i.e. the time at which the particles reach an approximate depth of $1000\ m$ for each value of $v_s$) are displayed in Fig. 5, both as a function of $v_s$ and of $t_f$. Similarly, the values of $r_{h,v}^{(co)}$ are presented in Fig. 6. The behavior can be understood as resulting from two factors: on the one hand smaller $v_s$ requires larger $t_f$ to reach the final depth, and larger integration time $t_f$ allows for accumulation of larger differences between

trajectories. On the other hand the Coriolis and inertial terms in Eqs. (10)-(11) are proportional to $\tau_p(\beta - 1) = v_s/g$ so that their magnitude decreases for smaller $v_s$. The combination of these two competing effects shapes the curves in Figs. 5 and 6, which for the vertical-difference case turn-out to be non-monotonic in $v_s$ or $t_f$.

     In all cases, the differences (both in vertical and horizontal) between the simple dynamics (9) and the corrected ones in Eqs. (10) and (11) are negligible when compared with typical particle displacements, or even with model grid sizes. For example,

we imposed in our simulations a vertical displacement close to 1000 m, whereas the mean root square difference with respect to simple sinking is below $1\ m$ for the Coriolis case (Fig. 5) and below $1\ cm$ for the inertial case (Fig. 6). In the horizontal direction, displacements during those times are of the order of hundreds of $km$, whereas the corrections introduced by the Coriolis and inertial terms are in the worst cases of the order of a few kilometers or of tens of meters, respectively. In particular, the most important impact (horizontal differences of tens of kilometer) is attributed to the Coriolis term for particles sinking

at 5 m/day (Fig. 5). This indicates that the inclusion of the Coriolis term would be required to properly model slowly sinking particles at high latitudes. It is worth noting that although the small value of Rossby number $\simeq 0.01$ for mesoscale processes might indicate a strong influence of the Coriolis force in Eq. (8), its influence on particle dynamics becomes negligible because it is multiplied by $\tau_p$ or equivalently, the Stokes number, which is significantly small for biogenic particles. Nevertheless Rossby number coincides with the ratio of inertial term to Coriolis term in Eq. (8) and its value $\simeq 0.01$ explains the difference

of two orders of magnitude among the corrections arising from the inertial force and from Coriolis. The trajectories of the full dynamics ruled by Eq. (8) are nearly identical to the ones under the approximation which keeps only the sinking term and Coriolis, so that the corresponding comparison to $\mathbf{r}_i^{(0)}$ gives a figure essentially identical to Fig. 5 (not shown).

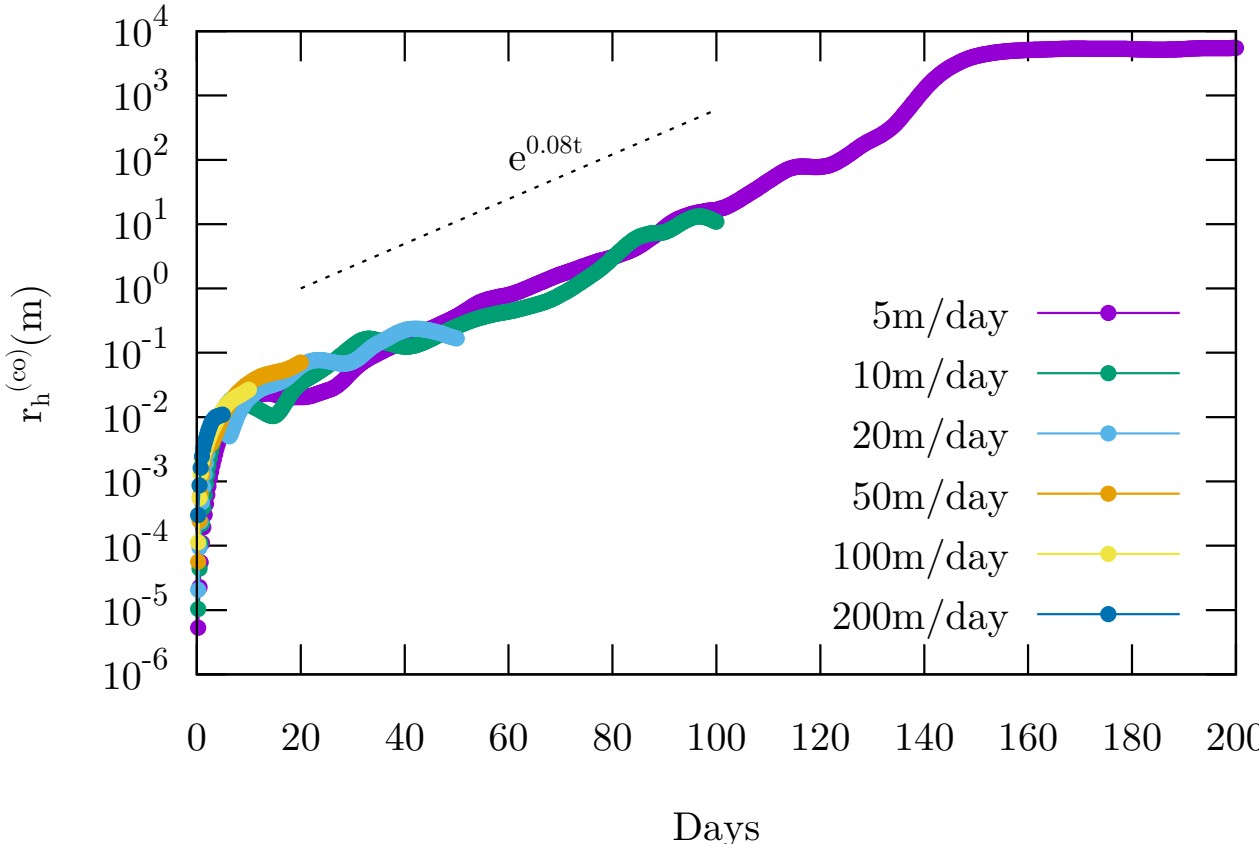

**Figure 4.** Root mean square difference per particle, as a function of time, between horizontal particle positions computed with Eq. (9) and with Eq. (10), i.e. with and without the Coriolis term. The different colors correspond to distinct values of the unperturbed sinking velocity. The dashed line is an exponential with slope $0.08\ day^{-1}$.

In summary, for the range of sizes and densities of the marine particles considered here, the sinking dynamics is essentially given by the velocity $\mathbf{v} = \mathbf{u} + \mathbf{v}_s$, which has been the one used in some oceanographic studies (Siegel and Deuser, 1997; Siegel et al., 2008; Roullier et al., 2014). Note however that a new question arises: what is then the reason for the observed clustering of falling particles (Logan and Wilkinson, 1990; Buesseler et al., 2007; Mitchell et al., 2008)? The argument of the non-inertial dynamics of the particles does not serve since $\nabla \cdot \mathbf{v} = \nabla \cdot \mathbf{u} = 0$. A possible response is explored in the next section.

## 5 Geometric clustering of particles

Compressibility of the particle-velocity field, i.e. $\nabla \cdot \mathbf{v} \neq 0$, which can arise from inertial effects even when the corresponding fluid-velocity field is incompressible $\nabla \cdot \mathbf{u} = 0$, has been identified as one of the mechanisms leading to preferential clustering

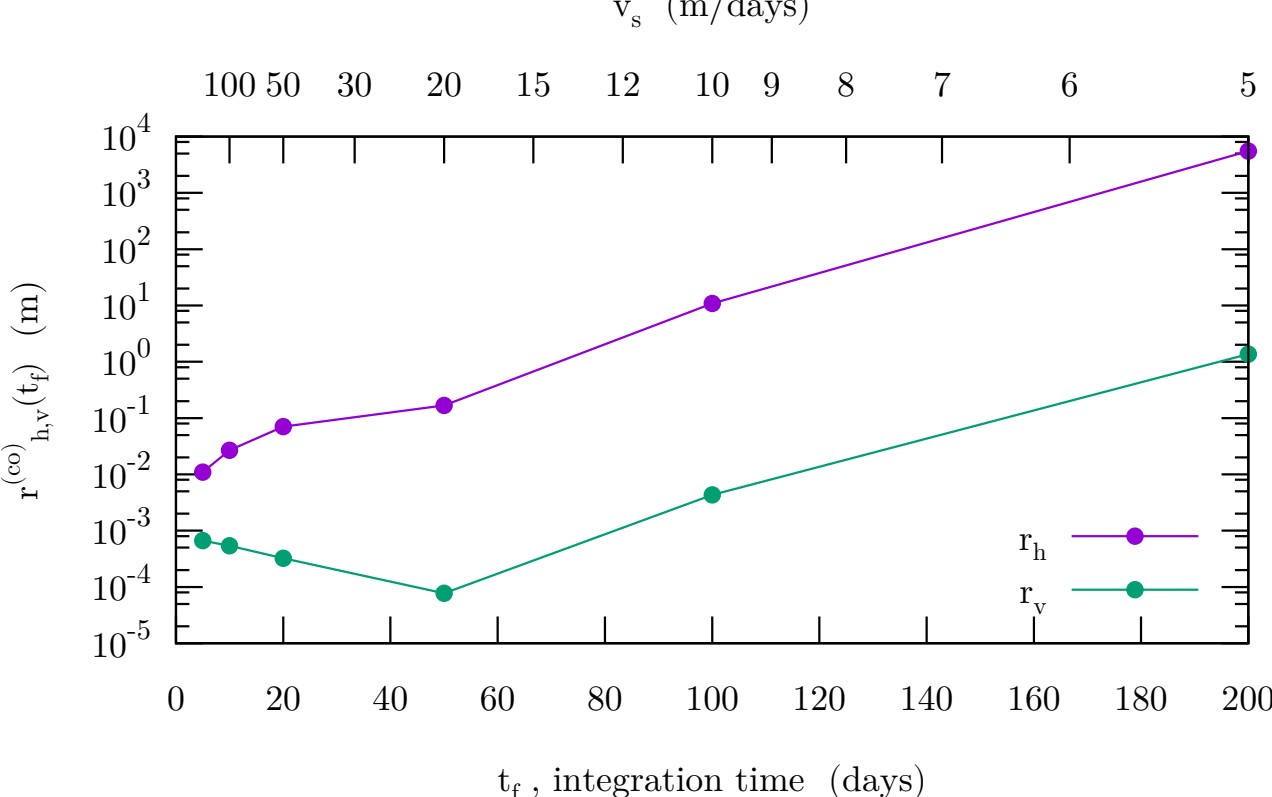

**Figure 5.** Root mean square difference per particle between final positions (at times $t_f$ stated in Table 2) computed with and without the Coriolis term (Eqs. (10) and (9), respectively). Data are presented as a function of the unperturbed sinking velocity $v_s$ used (upper horizontal scale) and of the final integration time $t_f$ (lower horizontal scale). Upper violet line, the horizontal difference $r_h^{(co)}(t_f)$; lower green line, the vertical difference $r_v^{(co)}(t_f)$.

of particles in flows (Squires and Eaton, 1991; Balkovsky et al., 2001). This is so because $\rho(t)$, the particle density at time $t$ at the location $\mathbf{r} = \mathbf{r}(\mathbf{r}_0, t)$ of a particle that started at $\mathbf{r}_0$ at time zero, satisfies $\rho(t) = \rho(0)\delta^{-1}$, where $\delta$ is a dilation factor equal to the determinant of the Jacobian $|\frac{\partial \mathbf{r}}{\partial \mathbf{r}_0}|$, which satisfies

$$\frac{1}{\delta}\frac{D\delta}{Dt} = \nabla \cdot \mathbf{v} \tag{14}$$

5  or, using $\delta(0) = 1$:

$$\delta(t_f) = e^{\int_0^{t_f} dt \nabla \cdot \mathbf{v}} . \tag{15}$$

Thus, particles will accumulate (i.e. higher $\rho(t_f)$) in final deep locations receiving particles whose trajectories have predominantly travelled through regions with $\nabla \cdot \mathbf{v} < 0$. We have seen however that, to a good approximation $\nabla \cdot \mathbf{v} \approx \nabla \cdot \mathbf{u} = 0$ since

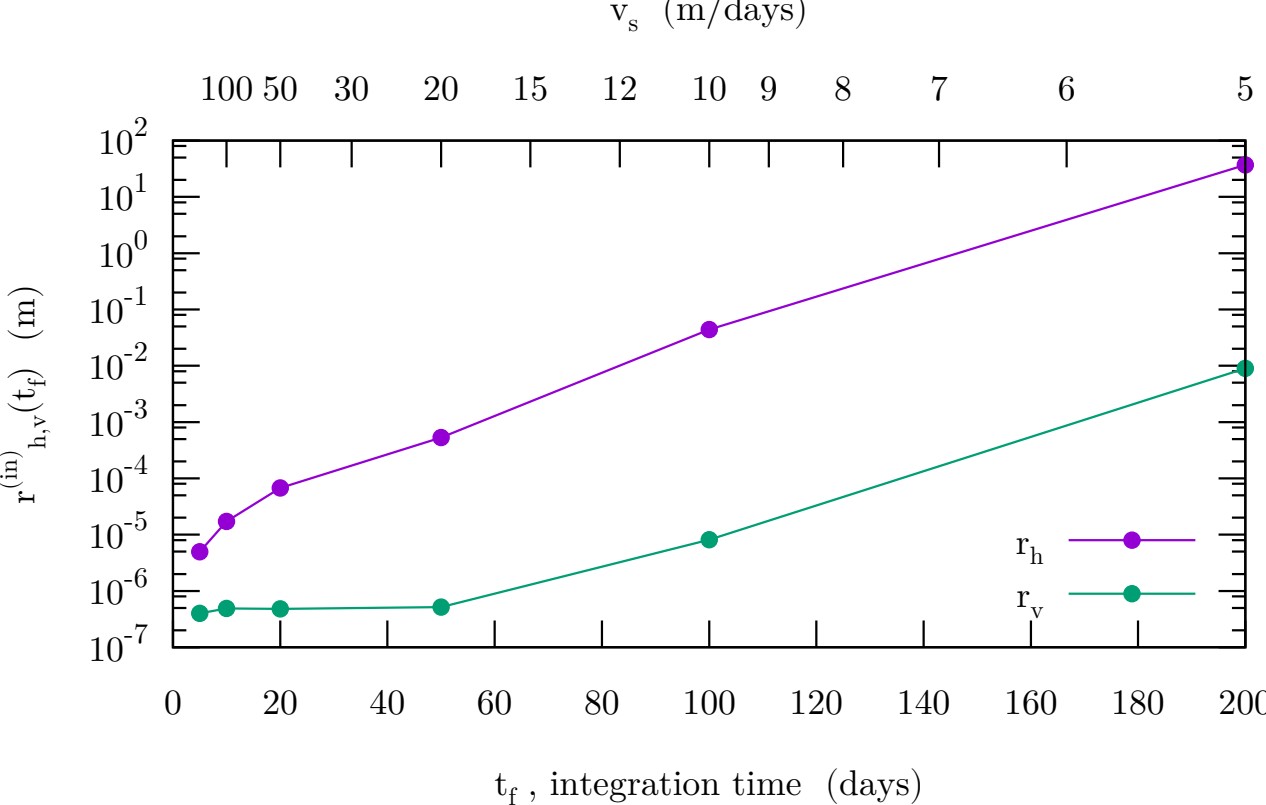

**Figure 6.** Root mean square difference per particle between final positions (at times $t_f$ stated in Table 2) computed with and without the inertial term (Eqs. (11) and (9), respectively). Data are presented as a function of the unperturbed sinking velocity $v_s$ used (upper horizontal scale) and of the final integration time $t_f$ (lower horizontal scale). Upper violet line, the horizontal difference $r_h^{(in)}(t_f)$; lower green line, the vertical difference $r_v^{(in)}(t_f)$.

inertial effects can be neglected for the type of marine particles we consider here, and then the three-dimensional particle-velocity field is incompressible.

We now reproduce numerically a typical situation in which clustering of marine particles is observed. We release particles uniformly in an horizontal layer close to the surface, we let them sink within the oceanic flow and we finally observe observe the distribution of the locations where they touch another horizontal deeper layer. The domain chosen is the rectangle $12°S$ to $35°S$ and $4°E$ to $19°E$ (orange rectangle in Fig. 3). We divide the domain horizontally in squares of side $1/25°$, then initialize 1000 particles at random positions in each of them in August 20, 2008 at depth $z = -100\ m$ (i.e. the bottom of the euphotic layer, starting point of our biogenic particles), and then integrate each trajectory until it reaches $-1000m$ depth. We use expression (9) for the velocity, with $v_s = 50m/day$. In order to avoid any small fluctuating compressibility arising from

the noise term we put $\mathbf{W} = \mathbf{0}$ but we have checked that the result in the presence of noise is virtually indistinguishable (not shown). At the bottom layer ($z = -1000\ m$) we count how many particles arrive to each of the $1/25°$ boxes and display the result in Fig. 7(a). Despite $\nabla \cdot \mathbf{v} = 0$ we see clear preferential clustering of particles in some regions related to eddies and filaments. We note that our horizontal boxes have a latitude-dependent area so that distributing particles at random in them produces a latitude-dependent initial density which could lead to some final inhomogeneities. We have checked however that for the range of displacements of the particles, this effect is everywhere smaller than $5\%$ and thus can not be responsible for the large clustering observed in Fig. 7(a). Nevertheless, this effect will be taken into account later.

We explain the observed particle clustering by considering the field displayed in Fig. 7(a) as a projection in two dimensions of a density field (the cloud of sinking particles) which evolves in three-dimensions. Even if the three-dimensional divergence is zero, and then an homogeneous three-dimensional density will remain homogeneous, a two-dimensional cut or projection can be strongly inhomogeneous. This mechanism has been proposed to explain clustering and inhomogeneities in the ocean surface (Huntley et al., 2015; Jacobs et al., 2016), but we show here that it is also relevant for the crossing of a horizontal layer by a set of falling particles.

A simple way to confirm that this clustering arises from the two-dimensionality of the measurement is to estimate the changes in the horizontal density of evolving particle layers as if they were produced just by the horizontal part of the velocity field. This is only correct if an initially horizontal particle layer remains always horizontal remains always horizontal during the sinking process, which is not true. But, given the huge differences in the values of the horizontal and vertical velocities in the ocean, we expect this approximation to capture the essential physics and provide a qualitative explanation of the observed cluster. We expect the approximation to become better for increasing $v_s$, because of the shorter sinking time during which vertical deformations could develop. Thus we compute the two-dimensional version of the dilation field, $\delta_h(\mathbf{x}, t_f)$, at each horizontal location $\mathbf{x}$ in the deep layer at $z = -1000m$:

$$\delta_h(\mathbf{x}, t_f) = e^{\int_0^{t_f} dt\, \nabla_h \cdot \mathbf{v}} \tag{16}$$

with the horizontal divergence

$$\nabla_h \cdot \mathbf{v} \equiv \frac{\partial v_x}{\partial x} + \frac{\partial v_y}{\partial y} = \frac{\partial u}{\partial x} + \frac{\partial v}{\partial y} = -\frac{\partial w}{\partial z} \ , \tag{17}$$

where in the second equality we have used Eq. (9) from which $\nabla_h \cdot \mathbf{v} = \nabla_h \cdot \mathbf{u}$ and the third one is a consequence of $\nabla \cdot \mathbf{u} = 0$. In order to get the values of $\delta_h$ on a uniform grid on the $-1000m$ depth layer at the arrival date $t_f$ of the particles in the previous simulation, we integrate backwards in time trajectories from grid points separated $1/50°$ at $z = -1000m$ until they reach $-100m$. The starting date ($t_f$) of the backwards integration was September 7, 2008, i.e. 18 days after the release date used in the previous clustering experiment. This value correspond to the average duration time of trajectories in that experiment. Then $\delta_h$ was computed integrating in time the values of $\nabla_h \cdot \mathbf{v}$ along every trajectory using Eq. (16).

Figure 7(b) displays the quantity $\delta(\mathbf{x}, t_f)^{-1} \cos(\theta_f)/\cos(\theta_0)$, which gives the ratio between densities in the upper and lower layer, corrected with the angular factors controlling the area of the horizontal boxes so that this can be compared with the ratio between particle numbers displayed in Fig. 7(a). $\theta_f$ is the latitude of point $\mathbf{x}$, and $\theta_0$ is the latitude of the corresponding

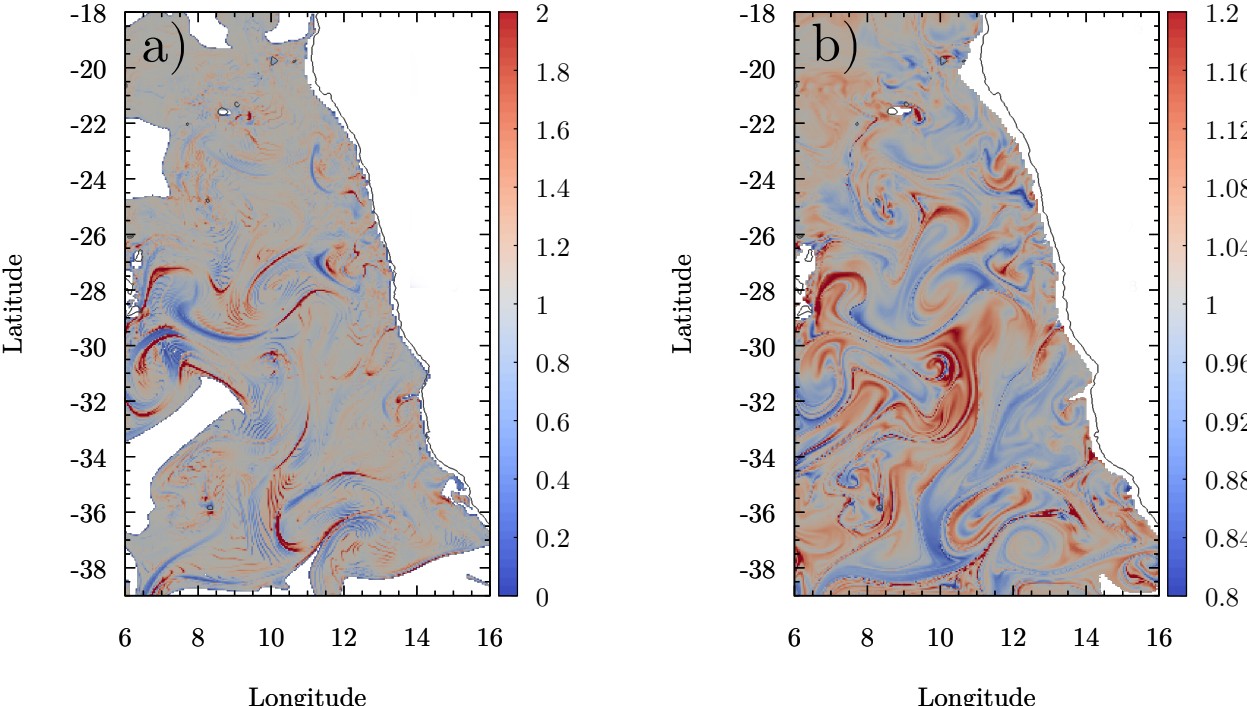

**Figure 7.** Results of the clustering numerical experiments of Sect. 5. a) $N_f/N_0$, the number of particles $N_f$ arriving to an horizontal box of size $1/25°$ in the horizontal layer at $z = -1000\ m$, normalized by the number of particles $N_0 = 1000$ released from the upper $z = -100\ m$ layer. b) The corrected dilation factor $\delta(\mathbf{x}, t_f)^{-1} \cos(\theta_f)/\cos(\theta_0)$ mapped on the final $z = -1000\ m$ layer. It gives the ratio between horizontal densities at the final and initial locations, corrected with the latitudinal dependence of the horizontal boxes used in panel a), to give an estimation of the local particle number ratio between lower and upper layer. Black thin line represents the coastline; white oceanic areas indicate in a) regions which do not receive any falling particles; in b) regions from which the backward integration ends up outside the domain.

trajectory in the upper $z = -100\ m$ layer. As stated before, the latitudinal corrections by the cosine terms are always smaller than a $5\%$. Although there is no perfect quantitative agreement, there is clear correspondence between the main clustered structures in panels (a) and (b) of Fig. 7, confirming that they originate from the horizontal dynamics in an incompressible three-dimensional velocity field. We have checked in specific cases that locations with larger differences between Figs. 7(a) and (b) correspond to places with large dispersion in the arrival times to the bottom layer, indicating deviations from the horizontality assumption.

## 6 Conclusions

We have studied the problem of sinking particles in a realistic oceanic flow, focussing in the range of sizes and densities appropriate for marine biogenic particles. Starting from a modeling approach in terms of the MRG equation (1), our conclusion is that the simplest approximation given by Eq. (9) in which particles move passively with the fluid flow with an added constant settling velocity in the vertical direction is an accurate framework to describe the sinking process in the type of flows and particles considered. A re-assessment of these assumptions may be required if more complex processes (such as aggregation/disaggregation) are included and when super-high resolution (submesoscale and below) mimicking the real ocean will become available.

Corrections arising from the Coriolis force turn out to be about 100 times larger than the ones coming from inertial effects, in agreement with the results in Sapsis and Haller (2009) or in Beron-Vera et al. (2015), but both of them are negligible when compared to the effects of passive transport by the fluid velocity plus the added gravity term, except for very slowly sinking particles in high latitudes.

If the fluid flow field $\mathbf{u}(\mathbf{r}, t)$ has vanishing divergence then the same is true for the particle velocity field defined by the approximation in Eq. (9). Then, no three-dimensional clustering can occur within this approximation. Nevertheless, we have shown that two-dimensional cuts or projections of evolving three-dimensional particle clouds display horizontal clustering.

*Competing interests.* The authors declare that they have no conflict of interest.

*Acknowledgements.* We acknowledge support from the LAOP project, CTM2015-66407-P (AEI/FEDER, EU), from the Office of Naval Research Grant No. N00014-16-1-2492, and through a Juan de la Cierva Incorporación fellowship (IJCI-2014-22343) granted to V.R.

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
