# Peer review of "Modeling the dynamical sinking of biogenic particles in oceanic flow"

_Nonlinear Processes in Geophysics, 2016_

## Referee Comment (RC1) · Anonymous Referee #1 · 13 Jan 2017

SUMMARY: The manuscript "Modeling the dynamical sinking of biologenic particles in oceanic flow" presents an interesting study of particle sedimentation in realistic oceanic flows. In particular, it focuses in a simplified dynamics accounting for gravitational settling and also scrutinizes the effect of additional terms such as that due to Coriolis force (accounting for rotation) and inertial terms. The two latter effects are anyhow taken into account only perturbatively in Stokes as the authors mainly focus in the regime of small St number, which is shown (in Sect. 2, which is a nice review on particulate matter in the ocean) to be relevant to a wide range of sediments in the ocean. The manuscript presents numerical simulation of particles advected in Regional Ocean Modelling System simulating the Benguela region. In order to account for the unresolved scales, particles dynamics is also supplemented by a noise term with horizontal and vertical diffusivities appropriately chosen. The main results of the manuscript are as follows:

[Figure]

(1) it is shown that both the vertical and horizontal displacement are well captured by the gravitational settling term, $v_s$, with a negligible (at high $v_s$) of the Coriolis and inertial term, this appears to be true both for the average displacement and for individual particles trajectories; (2) even considering only the presence of settling speed the particles manifest the presence of clustering in the horizontal dynamics. As for the latter the authors argue that the standard explanation based on particle velocity compressibility cannot hold as the effective (three-dimensional) velocity field is incompressible as the settling speed is constant. However, the authors show that under the hypothesis of constant settling the horizontal dynamics would be effectively controlled by a 2d flow which is compressible, and provide some test, based on integrating the particle density field in such compressible 2d flow, showing that this mechanism can at least partially explain the observed clustering.

GENERAL COMMENTS I found the manuscript interesting and, in general, very well written. In particular, it offers a very nice review on particulate matter in oceanic flows and also on the limit of validity of models for particle dynamics. The presented results are quite interesting and I think relevant both in modeling and interpretation of data. However, though the presented results are quite sounding, I think the paper would benefit of some more specifications in a few points to make clearer the results and their applicability. For this reason here below I expand a short list of specific comments that the authors can use in revising their manuscript.

SPECIFIC COMMENTS

- I think the authors should specify whether eqs 9,10,11 are evolved using the same realization of noise or not. I mean when considering the effect of the different term the comparison is done using the same realization of the noise ? If not what is the impact on the displacement due only to noise?

- Maybe I'm missing something, while I understand that the mean displacement does not depend much on the various term (Coriolis and inertial term) I find a little surprising

the fact that also individual displacement seems to be poorly sensitive. The reason is as follows. I do expect the particle dynamics to be chaotic (correct me if I'm wrong) consequently the presence of different term on the dynamics (assuming same noise, otherwise even the simple presence of noise would produce the same effect) should cause at least a small displacement that it is then amplified by chaos, so I cannot properly understand why this effect is not seen. Can the authors please comment on this?

- Fig.5 shows that the effect of Coriolis forces becomes more important for small $v\_s$, moreover the curve seems to be non-monotonic, especially for the vertical displacement it appears like if there is a sort of minimum at $v\_s \approx 20$ m/s. Could the authors comment on these aspects.

- The explanation proposed by the authors for understanding particle clustering is quite sounding and interesting. It would be nice if the authors could compare and comment their explanation with that provided in Bec, et al Phys. Rev. Lett., 112, 184 501 and also in K Gustavsson, et al. "Clustering of particles falling in a turbulent flow" Phys. Rev. Lett., 112, 214501 (2014). Essentially the authors argue that in the limit of large St (i.e. for large settling velocities) the particles fall rapidly with respect to the characteristic time of the flow so that effectively is like the flow becomes "delta-correlated" so that inertial dissipative dynamics becomes responsible for clustering instead of the compressibility effect typical of small St. How does clustering depends on $v\_s$ here? If I understand the argument by the authors the larger $v\_s$ the more appropriate becomes the approximation of vertical shift, then I think the idea of the fluid becoming "delta-correlated" should apply also here and so the dynamics become as in a compressible 2d and delta-correlated flow. Is this correct? if yes less effective clustering should be present for small $v\_s$. Please comment.

- Stratification: I think the authors should specify whether the model used consider stratification or not. In general stratification is present in the ocean and it may impact sensibly particle dynamics (especially when $\beta$ is not too far from 1) and, in

particular, particle clustering, for a recent study in this direction the authors may refer to

A. Sozza et al "Large scale confinement and small-scale clustering of floating particles in stratified turbulence", Phys. Rev. Fluids 1, 052401 (2016)
* * *

---

## Referee Comment (RC2) · Anonymous Referee #2 · 1 Feb 2017

Report on the manuscript "Modeling the dynamical sinking of biogenic particles in oceanic flow" by Pedro Monroy, Emilio Hernandez-García, Vincent Rossi, and Cristobal Lopez

SUMMARY

The paper discusses the problem of modelling the dynamics of small, biogenic particles in the ocean, with a specific attention to the sinking motion.

The authors focus on the behaviour of small organisms whose dynamics can be described by using the Maxey-Riley-Gatignol equation for the unsteady particle motion in creeping flow conditions. Particles are assumed to be point-like, with spherical symmetry. The main goal of the paper is to assess the relative importance, in comparison to the combined action of ocean passive transport and gravity, of the Coriolis and inertial

forces.

Starting from the equation of motion in a general flow (neglecting Faxen corrections and Basset history terms) : first the authors discuss how this eq. is modified in the presence of an underlying rotating fluid and get to the dynamical eq. (5), then they discuss how this last can be simplified, with different degrees of approximation, in the limit of very small inertia, St«1, and get to the kinematic eqs. 9-10-11.

The simplified motion kinematic equations are then numerically integrated for particles released in a realistic ocean flow, described by daily averaged currents and vertical velocities obtained from ROMS (Regional Ocean Modelling System) simulation of the Benguela region (Angola).

On the basis of the numerical experiments, the authors conclude that for the sinking motion of the small biogenic particles here analysed, the important ingredients are : the passive advection due to the ocean mesoscale currents and the resolved vertical motion, the action of gravity, plus the action of some eddy diffusivity to describe the effect of unresolved motions.

Finally, the authors comment on the biogenic particle clustering in a 3D incompressible ocean, by estimating the changes in the horizontal density of evolving particle layers moving with the horizontal part of the velocity field only.

GENERAL COMMENTS AND RECOMMENDATION

The manuscript is well-written and the topic is of clear interest both for the physical and biological oceanography communities. However, as it is now, the results seem to me not sufficiently analysed to draw the actual conclusions contained in the manuscript. Hence, in reason of the specific comments reported below, I can not at present recommend publication. In the following I summarise a series of concerns that, I think, the author should consider to improve the manuscript quality and clarity.

SPECIFIC COMMENTS

1) Abstract The sentence "Our aim is to unify the theoretical investigations with its applications in the oceanographic context and considering a mesoscale simulation of the oceanic velocity field." sound to me a bit too ambitious for what is really done in the paper. In the end the authors never really integrate the particles motion via their dynamical equations.

Also, the sentence "By using the equation of motion of small particles in a fluid flow, we assess the influence of physical processes such as the 5 Coriolis force and the inertia of the particles" is a bit exaggerated. Rather then assessing, I think that the authors estimate the influence of physical processes, on the basis of some simplifying assumptions.

2) In the introduction the same kind of sentences is used. "We assess the influence of physical processes such as the Coriolis force and the inertia of the particles with respect to the settling velocity. We also study the spatial distribution of particles falling onto a plane of constant depth above the seabed and we observe clustering of particles that is interpreted with simple geometrical arguments which do not require physical phenomena beyond passive transport and constant terminal velocity."

I think that these sentences should be somehow smoothed, since the statistical analysis here reported on both the contributions of different forces in the particles dynamics, and the clustering are interesting, but rather simple.

3) Sect. 3.1

3a) A relevant reference about the importance of Basset history term is missing. The effect of the Basset history force on particle clustering in homogeneous and isotropic turbulence Phys. Fluids 26 (2014).

3b) Page 6, line 10: a discussion here starts about the validity of MRG eq. and the specific ocean conditions. I find this discussion too vague. Authors should be more carefully their working hipothesis. In particular: Kolmogorov scales (spatial and temporal) are based on the kinetic energy dissipation rate, \epsilon. This can change a lot i the ocean, both horizontally, and vertically. So how do the authors choose the adopted value (which should be \epsilon= 1.e-6 m^2/s^3)? Have they obtained it from ROMS stat of the Benguela region? Does it vary vertically and /or horizontally? If \epsilon= 1.e-6 m^2/s^3, then \eta=1mm and \tau_{eta}=1s : are these the values used by the authors to assess MRG eq. validity?

4) Sect.3.2 4a)There are two references about inertial particles in rotation flows that the authors seem to miss. These are: Dynamics of Particles Advected by Fast Rotating Turbulent Fluid Flow: Fluctuations and Large-Scale Structures, Phys Rev Lett 14 (1998). Coherent Structures and Extreme Events in Rotating Multiphase Turbulent Flows, Phys Rev X 6 (2016).

4b) In particular in the latter, it is discussed, in addition to the Stokes drag and added mass terms, the role of centrifugal and Coriolis forces onto the dynamics and dispersion of inertial particles, by using DNS at high resolution. Authors should consider their assumptions - and conclusions- in view of these results.

The Rossby number for the particle turbulent dynamics matters and clearly distinguishes cases when rotation related effects are sub-dominant from cases when rotation dominates both the flow and the inertial particles dynamics. E.g. it seems that if Ro=(\epsilon k_f^2)^1/3/\Omega with k_f = 2.pi/10km —> Ro= 1; if k_f = 2.pi/100km —> Ro=0.2 and rotation might matter. Although in the ocean these turbulent parameters might not be the relevant ones and considering also that for the biogenic particles the Stokes number are very small, the authors should better justify their choices.

5) Sect. 4 5a)The authors consider daily averages of the ocean currents and vertical velocity calculated from ROMS. This implies that not only the dynamics at spatial scales of the order or smaller that the grid size is lost, but also the temporal variability. The authors choice is to use a white noise to the flow velocity to account for the huge gap of scales existing between 200 microns and about 10km (and similarly for the time

scales). This is clearly a crude approximation, in view of the fact that there is a consistent literature showing the role of such sub-grid motions in the Lagrangian dynamics of particles, by using stochastic or kinematic closures (see Lagrangian stochastic modeling in coastal oceanography, J. Atmos. Oceanic Technol., 19 (2002), and Lagrangian simulations and interannual variability of anchovy egg and larva dispersal in the Sicily Channel, J. Geophys. Res. Oceans, 119, 2014).

Probably it is justified for deep stratified waters, but this has to be discussed.

5b) Also, how are the eddy diffusivity values (or and vert) chosen? Are these extracted from ROMS simulations in the Benguela region? Before analysing simulations of eqs 11 or 10 compared to eq. 9, the authors should show what is the effect of the noise.

So it is crucial to see first how the trajectories are modified by considering the advection by the ROMS velocity only, or the advection by the ROMS velocity and the noise term. Ans also estimate in these cases the mean traveled distance. If the noise contribution is an order (or more) of magnitude larger that inertia or rotation contributions, then we might question the choice of applying these noise terms as representatives of the dynamics at all unresolved scales..

5c) How is exactly estimate the inertial term in eq. 11? The appropriate choice would be to estimate it from the true ROMS simulations and not from the daily averaged snapshots, since these last never had the correct temporal variability. Indeed we could make a crude approximation to compare the inertia and Coriolis force: the former is the resolved velocity divided by the time scale between one snapshot and the other, $\sim 1/T = 1.15e-5/s$, while the latter is the resolved velocity times $2*Omega=14.5e-5/s$. So this already tells us that the Coriolis term will be more important...But this is due to the way forces are calculated here.

5d) A convenient and more informative choice to estimate the importance of different terms would be to plot the horizontal and vertical root-mean-square error growth between different models trajectories, as a function of time. Table 2 is indeed not really

informative.

6) Sect. 5 I find this section highly speculative and not very informative. It is clear that a 2D cut of a 3D incompressible field will be compressible, and hence exhibit some sort of clustering. But unless the analysis becomes more specific, I think the authors should remove this section and keep the comments there contained for the conclusions. Similarly, the abstract should be rephrased.

MINOR COMMENTS

i) it seems to me that there is a incorrect sign in the expression of the modified terminal velocity at page 7, line 15. it should be v'= (1-\beta)[g - \Omega \times \Omega \times r]; moreover it should be emphasized that the "r" appearing in the centrifugal force expression is the distance from the rotation axis. So unless this last is zero, this is not correct either.

ii) label of figs. 4 and 5 seem to me wrong. The dimension is not m/s but (m/day), I guess. Similarly, at page 11 line 21, v_s should be v_s= 5m/day and not 5 m/s. Please check this throughout the paper.

---

## Author Comment (AC1) · 30 Mar 2017

Response to Referee 1:

We acknowledge Referee 1 for the careful reading and constructive comments. Referee 1 qualifies the paper as **'interesting and, in general, very well written'** and the results **'are quite interesting and I think relevant both in modeling and interpretation of the data'**. However, he/she presents a list of specific comments that we have addressed through our detailed responses below, together with specific changes made in the manuscript. Page, figure and line numbers refer to the revised version of the manuscript.

**- I think the authors should specify whether eqs 9,10,11 are evolved using the same realization of noise or not. I mean when considering the effect of the dif-**

**ferent term the comparison is done using the same realization of the noise? If not what is the impact on the displacement due only to noise?**

All the simulations were performed using the same realization of noise. Although this was already stated in the previous version, we have reworded the corresponding sentence to make it clearer (page 11, paragraph after Eq. (11)): *"We use in each case identical initial conditions and the same sequence of random numbers for the noise terms. In this way we guarantee that any difference in particle trajectories arises from the inclusion or not of the inertial and Coriolis terms in Eqs. (8)-(11)."*

As additional information we show in Figures 1 and 2 in this Response letter the influence of noise on horizontal and vertical displacements versus time, respectively. We have run Eq. (9) with and without noise for a set of N=6000 particles and have computed the root mean square horizontal and vertical distances between the two cases as a function of time (and for different values of settling velocity). Noise-induced differences are larger than the ones induced by the Coriolis and inertial ones (Figs. 4 and 6 of the manuscript). Therefore the use of different realizations of noise would prevent us to observe the (weak) influence of Coriolis and inertia.

**- Maybe I'm missing something, while I understand that the mean displacement does not depend much on the various term (Coriolis and inertial term) I find a little surprising the fact that also individual displacement seems to be poorly sensitive. The reason is as follows. I do expect the particle dynamics to be chaotic (correct me if I'm wrong) consequently the presence of different term on the dynamics (assuming same noise, otherwise even the simple presence of noise would produce the same effect) should cause at least a small displacement that it is then amplified by chaos, so I cannot properly understand why this effect is not seen. Can the authors please comment on this?**

The referee is completely right: the particle dynamics is chaotic. In the ocean, exponential growth of horizontal distances is observed up to scales of about 40 km (Poje et

al, 2010). Thus, it is expected to observe this exponential distance growth, assuming the same noise realization, when comparing trajectories with or without the inertial or Coriolis terms. Since this is an important issue, also requested by Referee 2, we have added a new Figure 4 to our manuscript. It shows the time evolution of the root mean square difference per particle between horizontal displacements computed from Eq. (9) and Eq. (10) in the text, that is, without and with Coriolis forces. Exponential growth with an exponent of about 0.08 days-1 is observed. Comparison between inertial and non-inertial dynamics (not shown in the paper, but shown in this Response letter as Fig. 3) gives the same exponential behavior and the same exponent, although with difference two orders of magnitude smaller. This exponent corresponds to the Lyapunov exponent, whose value is in the range of results obtained by Bettencourt et al. (2012) for the same region and model.

This is not contradictory with the statements in the paper about negligible effect of the Coriolis or inertial terms. For example, despite the exponential growth the largest horizontal differences attained (for the Coriolis case) at the largest times are still of only 1-10 km (as reported in Figs. 4 and 5), much smaller than typical horizontal displacements at these times (hundreds of km).

In addition to the new Fig. 4 in the manuscript, we have included a reference to Bettencourt et al. and some sentences that discuss about this (Sect. 4).

- Poje, A.C et al. (2010). Resolution dependent relative dispersion statistics in a hierarchy of ocean models. Ocean Modelling, 31, 36-50.

- Bettencourt; J.H. et al. (2012). Oceanic three dimensional Lagrangian Coherent Structures: A study of a mesoscale eddy in the Benguela upwelling region. Ocean Modelling 51, 73-83.

**- Fig.5 shows that the effect of Coriolis forces becomes more important for small $v_s$, moreover the curve seems to be non-monotonic, especially for the vertical displacement it appears like if there is a sort of minimum at $v_s \approx 20$ m/s. Could**

**the authors comment on these aspects.**

We have added the following explanation in Sect. 4 while Figs. 5 and 6 have been redrawn with scales to make this point clearer. *"The behavior can be understood as resulting from two factors: on the one hand smaller $v_s$ requires larger $t_f$ to reach the final depth, and larger integration time $t_f$ allows for accumulation of larger differences between trajectories. On the other hand the Coriolis and inertial terms in Eqs. (10)-(11) are proportional to $\tau_p(1 - \beta) = v_s/g$ so that their magnitude decreases for smaller $v_s$. The combination of these two competing effects shapes the curves in Figs. 5 and 6, which for the vertical-difference case turn-out to be non-monotonic in $v_s$ or $t_f$ ."*

**- The explanation proposed by the authors for understanding particle clustering is quite sounding and interesting. It would be nice if the authors could compare and comment their explanation with that provided in Bec, et al Phys. Rev. Lett., 112, 184 501 and also in K Gustavsson, et al. "Clustering of particles falling in a turbulent flow" Phys. Rev. Lett., 112, 214501 (2014). Essentially the authors argue that in the limit of large St (i.e. for large settling velocities) the particles fall rapidly with respect to the characteristic time of the flow so that effectively is like the flow becomes "delta-correlated" so that inertial dissipative dynamics becomes responsible for clustering instead of the compressibility effect typical of small St. How does clustering depends on $v_s$ here? If I understand the argument by the authors the larger $v_s$ the more appropriate becomes the approximation of vertical shift, then I think the idea of the fluid becoming "delta-correlated" should apply also here and so the dynamics become as in a compressible 2d and delta-correlated flow. Is this correct? if yes less effective clustering should be present for small $v_s$. Please comment.**

We recall that the focus of our paper is not on particle clustering but on quantifying and assessing the importance of different terms in the dynamics in the trajectories of particles whose characteristics are typical of biogenic particles observed in the ocean. We have included a last section with some comments on clustering just to answer a

natural question that arose during our investigation: If inertia and Coriolis are negligible, why are there observations of clustering for this type of particles in the ocean? We show that spatial inhomogeneities can arise simply by the geometric way in which measurements are done. We do not claim this is the only explanation, but it is certainly the simplest one.

In our discussion on clustering we consider complete absence of inertia, i.e. $St = 0$. Then the mechanisms described in the two papers mentioned by the referee, both based on the effect of inertia, i.e. finite St, are necessarily not operating in our results. More specifically:

- In [Bec, et al Phys. Rev. Lett., 112, 184501 (2014)] several asymptotic regimes are considered. The one mentioned by the referee is the case of St»1. In this situation fluid velocity can be approximated by a delta-correlated noise. Within this nice approximation the authors are able to obtain analytical results on the effect of inertia on particle clustering. This regime is exactly the opposite of our St=0. In our case particle adapts to the flow faster than any change in the flow field, so that a delta-correlation approximation is not appropriate to our situation.

- In [Gustavsson, et al. Phys. Rev. Lett., 112, 214501 (2014)] also several asymptotic regimes are considered. The one mentioned by the referee is that of large "gravity parameter" F. In this case inertial particles fall very fast and strong clustering may occur by a mechanism of multiplicative amplification of the inhomogeneities already induced by a finite St. Again, this mechanism is absent in our $St = 0$ case.

Since it was not the focus of our paper we did not study systematically how our geometric clustering mechanisms depends on $v_s$ (this is left for future work). Intuitively, the larger $v_s$ the better will be the approximation from which we derive Eq. (16), i.e that a horizontal surface remains horizontal under evolution. This does not directly imply having less or more clustering.

In the revised manuscript we now refer to Bec (2014) and Gustavsson (2014) at the end

of Sect. 3 as examples of clustering mechanisms arising from inertia. We have also added the sentence *"We expect the approximation to become better for increasing $v_s$, because of the shorter sinking time during which vertical deformations could develop"* just before the last sentence before Eq. (16).

**- Stratification: I think the authors should specify whether the model used consider stratification or not. In general stratification is present in the ocean and it may impact sensibly particle dynamics (especially when $\beta$ is not too far from 1) and, in particular, particle clustering, for a recent study in this direction the authors may refer to A. Sozza et al "Large scale confinement and small-scale clustering of floating particles in stratified turbulence", Phys. Rev. Fluids 1, 052401 (2016)**

Certainly, there is density stratification in the ROMS numerical simulations used in our study. But we do not think that this is a relevant mechanism of clustering for the range of densities of biogenic particles we are studying, which is approximately $1050 - 2700 kg/m^3$. The reason is that fluid density ranges between $1020 - 1030 kg/m^3$ for surface waters, with large values of the order $1045 - 1050 kg/m^3$ arising only in very deep waters (several kilometers depth). Indeed, we consider here a constant size and density for each particle along its downward course; this means that we neglect biogeochemical and (dis)aggregation processes that may occur in nature but that are currently poorly known (and overly complex to be modelled in our framework, e.g. Maggi et al. 2013).

The effect of stratification discussed in Sozza et al. appears when particle density equals water density, so that particles get confined vertically in the isopycnal surface given by the condition "density of particle = density of fluid", which cannot be fulfilled in the ranges we consider here.

In this revised version of the manuscript we briefly mention at the end of Sect. 2.2 the work of Sozza et al., and we mention the weak impact of stratification on the value of

$v_s$ in the paragraph following Eq. (11).

[Figure]

[Figure]

**Fig. 1.** Root mean square difference per particle, as a function of time, between horizontal particle positions computed with Eq. (9) with and without noise.

[Figure]

The plot shows $r_v(m)$ on the vertical axis (logarithmic, from $10^{-1}$ to $10^2$) versus Days on the horizontal axis (from 0 to 200).

Legend:
- 5m/day
- 10m/day
- 20m/day
- 50m/day
- 100m/day
- 200m/day

**Fig. 2.** Root mean square difference per particle, as a function of time, between vertical particle positions computed with Eq. (9) with and without noise.

[Figure]

**Fig. 3.** Root mean square difference per particle, as a function of time, between horizontal particle positions computed with Eq. (9) and with Eq. (11), i.e. with and without inertia.

---

## Author Comment (AC2) · 30 Mar 2017

Response referee 2:

We acknowledge Referee 2 for the careful reading and detailed comments. Referee 2 states that the manuscript **"is well-written and the topic is of clear interest both for the physical and biological oceanography communities"**. However, he/she gives a series of concerns that should be addressed to improve quality and clarity. We report in the following detailed point-by-point responses, together with the specific changes made in the manuscript. Page, figure and line numbers refer to the revised version of the manuscript.

**1) Abstract The sentence "Our aim is to unify the theoretical investigations with its applications in the oceanographic context and considering a mesoscale sim-**

**ulation of the oceanic velocity field." sound to me a bit too ambitious for what is really done in the paper. In the end the authors never really integrate the particles motion via their dynamical equations.**

**Also, the sentence "By using the equation of motion of small particles in a fluid flow, we assess the influence of physical processes such as the 5 Coriolis force and the inertia of the particles" is a bit exaggerated. Rather then assessing, I think that the authors estimate the influence of physical processes, on the basis of some simplifying assumptions.**

- The abstract has been edited to gain clarity. The sentences pointed out by the referee are now written as *"Our aim is to evaluate the relevance of theoretical results of finite size particle dynamics in their applications in the oceanographic context. By using a simplified equation of motion of small particles in a mesoscale simulation of the oceanic velocity field, we estimate the influence of physical processes such as . . ."*

**2) In the introduction the same kind of sentences is used. "We assess the influence of physical processes such as the Coriolis force and the inertia of the particles with respect to the settling velocity. We also study the spatial distribution of particles falling onto a plane of constant depth above the seabed and we observe clustering of particles that is interpreted with simple geometrical arguments which do not require physical phenomena beyond passive transport and constant terminal velocity."**

**I think that these sentences should be somehow smoothed, since the statistical analysis here reported on both the contributions of different forces in the particles dynamics, and the clustering are interesting, but rather simple.**

- The final paragraph of the introduction has been rewritten.

**3) Sect. 3.1**

**3a) A relevant reference about the importance of Basset history term is missing.**

**The effect of the Basset history force on particle clustering in homogeneous and isotropic turbulence Phys. Fluids 26 (2014)**

- We acknowledge Reviewer 2' suggestion: the reference by Olivieri et al is certainly relevant here. We have included in the revised version of the manuscript just before Eq. (2).

**3b) Page 6, line 10: a discussion here starts about the validity of MRG eq. and the specific ocean conditions. I find this discussion too vague. Authors should be more carefully their working hipothesis. In particular: Kolmogorov scales (spatial and temporal) are based on the kinetic energy dissipation rate, $\epsilon$. This can change a lot in the ocean, both horizontally, and vertically. So how do the authors choose the adopted value (which should be $\epsilon = 10^{-6}m2/s3$)? Have they obtained it from ROMS stat of the Benguela region? Does it vary vertically and /or horizontally? If $\epsilon = 10^{-6}m2/s3$, then $\eta = 1mm$ and $\tau_\eta = 1s$ : are these the values used by the authors to assess MRG eq. Validity?**

- These values are adopted from observational measurements in the ocean, and mainly taken from Jimenez (1997). Specifically, they correspond to wind driven turbulence at the surface. Certainly all these turbulent parameters vary in the horizontal, in the vertical, and in time, but to estimate the validity of the assumptions leading to MRG and its simplifications one only needs to consider worst-case situations for which the wind-driven surface values, in the open-ocean case, are good representatives (oceanic deep currents are sluggish and turbulence intensity at depth is typically weaker, except maybe in the vicinity of topographic features not considered here).

In the new version of manuscript we have clarified where these values are taken from, and we have also indicated to which $\epsilon$ do they correspond (in paragraph after Eq. (2)):*"Because of wind, turbulence intensity is typically larger at the ocean surface, with values of turbulent energy dissipation in the range $1 \cdot 10^{-6}m^2/s^3 < \epsilon < 3 \cdot 10^{-5}m^2/s^3$ (Jimenez,1997), than at depth. The first condition for the validity of the MRG equation*

*that Maxey and Riley discussed in their original paper (Maxey, 1983) is that the particles have to be much smaller than the typical length scale of variation of the flow. This means that for multiscale (turbulent) flows the radius of the particle $a$ has to be much smaller than the Kolmogorov scale $\eta$, which according to the previous values of $\epsilon$, is typically $0.3mm < \eta < 2mm$ in the ocean surface (Okubo 1971, Jimenez 1997). Note that we need only to consider worst-case situations for assessing the validity of the different approximations".*

**4) Sect.3.2 4a)There are two references about inertial particles in rotation flows that the authors seem to miss. These are: Dynamics of Particles Advected by Fast Rotating Turbulent Fluid Flow: Fluctuations and Large-Scale Structures, Phys Rev Lett 14 (1998). Coherent Structures and Extreme Events in Rotating Multiphase Turbulent Flows, Phys Rev X 6 (2016).**

- We now cite these references at the beginning of Sect. 3.2, where rotation is introduced.

**4b) In particular in the latter, it is discussed, in addition to the Stokes drag and added mass terms, the role of centrifugal and Coriolis forces onto the dynamics and dispersion of inertial particles, by using DNS at high resolution. Authors should consider their assumptions - and conclusions- in view of these results. The Rossby number for the particle turbulent dynamics matters and clearly distinguishes cases when rotation related effects are sub-dominant from cases when rotation dominates both the flow and the inertial particles dynamics. E.g. it seems that if $Ro = (\epsilon k_f 2)1/3/\Omega$ with $k_f = 2\pi/10km$ $Ro = 1$; if $k_f = 2.\pi/100km$ $Ro = 0.2$ and rotation might matter. Although in the ocean these turbulent parameters might not be the relevant ones and considering also that for the biogenic particles the Stokes number are very small, the authors should better justify their choices.**

- The turbulent parameters used in this reference are not the appropriate for sinking

of biogenic particles in the ocean. However it is true in our case that Coriolis is much more important than inertia (about a factor of 100). But still both effects are negligible compared with plain advection and sinking velocity. We have included the following discussion close to the end of Sect. 4: *"It is worth noting that although the small value of Rossby number 0.01 for mesoscale processes might indicate a strong influence of the Coriolis force in Eq. (8), its influence on particle dynamics becomes negligible because it is multiplied by $tau_p$ or equivalently, the Stokes number, which is significantly small for biogenic particles. Nevertheless Rossby number coincides with the ratio of inertial term to Coriolis term in Eq. (8) and its value 0.01 explains the difference of two orders of magnitude among the corrections arising from the inertial force and from Coriolis."*

**5) Sect. 4 5a)The authors consider daily averages of the ocean currents and vertical velocity calculated from ROMS. This implies that not only the dynamics at spatial scales of the order or smaller that the grid size is lost, but also the temporal variability.**

**The authors choice is to use a white noise to the flow velocity to account for the huge gap of scales existing between 200 microns and about 10km (and similarly for the time scales). This is clearly a crude approximation, in view of the fact that there is a consistent literature showing the role of such sub-grid motions in the Lagrangian dynamics of particles, by using stochastic or kinematic closures (see Lagrangian stochastic modeling in coastal oceanography, J. Atmos. Oceanic Technol., 19 (2002), and Lagrangian simulations and interannual variability of anchovy egg and larva dispersal in the Sicily Channel, J. Geophys. Res. Oceans, 119, 2014)**

- Yes, we agree with the referee that our ROMS data contains less temporal and spatial scales than the real ocean (this is acknowledged line 1-2 p. 18, conclusion sect.). We also agree with the referee that our representation in terms of a white noise is a crude approximation. But note that we are not analyzing the different ways to resolve subgrid motions, but studying the relevance of the different terms (Coriolis, inertia...) in the particle dynamics. The limit situation of sub-grid motion resolved by white noise is the one that introduces stronger temporal and spatial gradients, and then it serves well to our purposes: we expect that under more realistic sub-grid motions the impact of the corrections to simple sinking we evaluate (Coriolis, inertia, ...) would be still smaller than found here.

Nevertheless we agree with the referee on the convenience to mention more elaborated approaches to the modeling of sub-grid scales. We cite now just before Eq. (7) the two references suggested by the referee.

**5b) Also, how are the eddy diffusivity values (or and vert) chosen? Are these extracted from ROMS simulations in the Benguela region? Before analysing simulations of eqs 11 or 10 compared to eq. 9, the authors should show what is the effect of the noise. So it is crucial to see first how the trajectories are modified by considering the advection by the ROMS velocity only, or the advection by the ROMS velocity and the noise term. Ans also estimate in these cases the mean traveled distance. If the noise contribution is an order (or more) of magnitude larger that inertia or rotation contributions, then we might question the choice of applying these noise terms as representatives of the dynamics at all unresolved scales.**

Horizontal diffusivity is taken according to Okubo (1971) formula at the spatial scale of the model resolution. This is stated in the paragraph before Eq. (9). For the vertical diffusivity, which is orders of magnitude smaller, it is much more difficult to find systematic studies of its dependence with resolution and with depth. Estimating realistic mixing coefficients in the ocean is perhaps one of the most important, still unresolved, challenge facing physical oceanographers today. Because of that, we have taken an average value (constant with depth) from different bibliographical sources, as in Rossi et al. (2013), again with the idea of considering a rather generic situation. The difference in particle trajectories with and without noise could be large (see response to

first point of referee 1). This is why we keep the same sequence of random numbers when repeating our simulations in adding new terms (as stated before Eq. (11)). By using the same noise realization we can focus our primary objective, i.e. to compare the impact of adding or removing the Coriolis or the inertial terms. We have reworded the pertinent sentences to make this clearer here (page 11, paragraph after Eq. (11)): *"We use in each case identical initial conditions and the same sequence of random numbers for the noise terms. In this way we guarantee that any difference in particle trajectories arises from the inclusion or not of the inertial and Coriolis terms in Eqs. (8)-(11)."*

**5c) How is exactly estimate the inertial term in eq. 11? The appropriate choice would be to estimate it from the true ROMS simulations and not from the daily averaged snapshots, since these last never had the correct temporal variability. Indeed we could make a crude approximation to compare the inertia and Coriolis force: the former is the resolved velocity divided by the time scale between one snapshot and the other, 1/T = 1.15e-5/s, while the latter is the resolved velocity times 2*Omega=14.5e-5/s. So this already tells us that the Coriolis term will be more important. . But this is due to the way forces are calculated here.**

- We first note that our ROMS configuration uses climatological forcing. Thus, intraday variability is not really strong. But in any case its impact can be assessed qualitatively. We have followed Tang et al (2012) to estimate the term $Du/Dt$, under the assumption that unresolved small scales/times behave as an added random contribution. Thus we have for the flow velocity $u = u_{resolved} + noise$. As indicated in Tang et al, the inertial term gets a contribution Dnoise/Dt, but it turns out to be negligible since it is multiplied by the Stokes time. Nevertheless we agree with the conclusion of the referee that Coriolis is more relevant than inertia, as explained in the response to the point 4b.

**5d) A convenient and more informative choice to estimate the importance of different terms would be to plot the horizontal and vertical root-mean-square error growth between different models trajectories, as a function of time. Table**

**2 is indeed not really informative.**

- This is a very pertinent suggestion, also reported by referee 1. We have added a new Figure 4 showing the time-dependence of the mean root square differences between trajectories.

**6) Sect. 5 I find this section highly speculative and not very informative. It is clear that a 2D cut of a 3D incompressible field will be compressible, and hence exhibit some sort of clustering. But unless the analysis becomes more specific, I think the authors should remove this section and keep the comments there contained for the conclusions. Similarly, the abstract should be rephrased.**

In earlier drafts Sect. 5 was not present and we decided to include it since it was a simple way to answer the natural question that arose during our investigations: If there is not important impact of non-inertial dynamics in the sinking dynamics of biogenic particles, why clustering is observed? We show that spatial inhomogeneities can arise simply by the geometric way in which measurements are done. We do not claim this is the only explanation, but it is certainly the simplest one. Then we do not agree with the referee and we consider that our paper becomes more self-contained and useful to the community by keeping Sect. 5.

MINOR COMMENTS

**i) it seems to me that there is an incorrect sign in the expression of the modified terminal velocity at page 7, line 15. it should be $v' = (1 - \beta)[g - \Omega \times \Omega \times r]$; moreover it should be emphasized that the "r" appearing in the centrifugal force expression is the distance from the rotation axis. So unless this last is zero, this is not correct either.**

- Thanks for the correction, now the mistake is fixed and we include in the text (before Eq. (5)) a clear definition of the vector "r".

**ii) label of figs. 4 and 5 seem to me wrong. The dimension is not m/s but (m/day),**

**I guess. Similarly, at page 11 line 21, $v_s$ should be $v_s = 5m/day$ and not 5 $m/s$. Please check this throughout the paper.**

- We have fixed these mistakes. Thanks for pointing them out.

---

## Author Response (AR2)

**Response: Minor changes**

**Referee #1:**

*1) On p.12 at the end of the page the authors write:*
*"To assess the impact of the Coriolis and of the inertial effects we compare the positions r^(co)(t), and r^(in)(t) with the simpler dynamics r^(in)(t) for each time t. " If I understood correctly the second r^(in) should be a r^(0).*
*Please check.*

We have corrected this sentence:

"To asses the impact of the Coriolis and of the inertial effects we compare the positions $r_i^{(co)}(t)$ and $r_i^{(in)}(t)$ with the simpler dynamics Eq. (9) which gives $r_i^{(0)}(t)$ for each time t."

*2) I have still a remark on the issue of chaoticity. More than a remark is that I would like to understand better the point made by the authors.*

*As far as I understood the authors provide evidence with new fig.4 that the root mean square difference between the horizontal particle position computed by using the simple settling model Eq. (9) and the refined one including the Coriolis term grows exponentially in time. Moreover the comment that the exponential rate of about 0.08days is in agreement with the order of magnitude of the Lyapunov exponent calculated using the same ROMS velocity model and region.*

*This seems to agree with my comment expressed in the previous report.*

*Now the issue is the following. Suppose you compute the root mean square deviation between the horizontal position of particles that all follow the same simple dynamics (9) but considering two sets of particles with the positions that are initially slightly different, by a small amount similar to what would be the displacement induced by the Coriolis term, say in 1day.*
*What would be the behavior of the root mean square displacement?*
*Since the system is chaotic the root mean square will be growing exponentially with the Lyapunov exponent (I think) close to 0.08 days^{-1}. Therefore the net result would be a displacement after 180 days similar to that observed in fig.4.*

*In other terms my doubt can be restated with the following question. Is the effect of the Coriolis term the same as the effect of a small displacement of the simpler settling dynamics (9)? If yes I think it is difficult to conclude about that "the inclusion of the Coriolis term would be required to properly model slowly sinking particles at high latitudes".*

The referee has understood well our results, and the answer to last question is yes.

In fact it is a general result in chaotic systems that, after a transient time of the order of \lambda^{-1} (\lambda is the Lyapunov exponent), the effect of a perturbation on the dynamics could be well approximated by a change in the initial condition. This can be seen heuristically by thinking in the linearized evolution of the difference z between two dynamics: dz/dt \approx \lambda z + f(t), where f(t) is the difference between the two dynamics, here the Coriolis term. The solution is z(t)=exp(t\lambda)[z_0+\int_0^t exp(-s\lambda)f(s)ds]. For t>>\lambda^{-1} and if f(s) does not grow too fast, the upper limit in the integral can be approximated by \infty, so that the effect of the different dynamics f(t) is equivalent to an effective initial difference z_0 -> z_eff.

Our aim with the sentence "the inclusion of the Coriolis term would be required to properly model slowly sinking particles at high latitudes" was to stress that the effect of Coriolis (or equivalently, the change in this effective initial condition) is larger at high latitudes. But since this is clear from the results presented here, and to avoid further confusion, we have deleted this sentence in the revised version.